# Probabilistic, spinally-gated control of bladder pressure and autonomous micturition by Barrington's nucleus CRH neurons

Hiroki Ito[1,2†], Anna C Sales[1†*], Christopher H Fry[1], Anthony J Kanai[3], Marcus J Drake[1,4], Anthony E Pickering[1,5*]

[1]School of Physiology, Pharmacology and Neuroscience, Faculty of Life Sciences, University of Bristol, Bristol, United Kingdom; [2]Department of Urology, Yokohama City University Graduate School of Medicine, Yokohama, Japan; [3]Department of Medicine and Pharmacology & Chemical Biology, University of Pittsburgh, Pittsburgh, United States; [4]Bristol Urology Institute, Bristol Medical School, University of Bristol, Bristol, United Kingdom; [5]Anaesthetic, Pain and Critical Care research group, Translational Health Sciences, Bristol Medical School, University of Bristol, Bristol, United Kingdom

**\*For correspondence:**
anna.sales@bristol.ac.uk (ACS);
tony.pickering@bristol.ac.uk (AEP)

[†]These authors contributed equally to this work

**Competing interests:** The authors declare that no competing interests exist.

**Abstract** Micturition requires precise control of bladder and urethral sphincter via parasympathetic, sympathetic and somatic motoneurons. This involves a spino-bulbospinal control circuit incorporating Barrington's nucleus in the pons (Barr). Ponto-spinal glutamatergic neurons that express corticotrophin-releasing hormone (CRH) form one of the largest Barr cell populations. Barr[CRH] neurons can generate bladder contractions, but it is unknown whether they act as a simple switch or provide a high-fidelity pre-parasympathetic motor drive and whether their activation can actually trigger voids. Combined opto- and chemo-genetic manipulations along with multisite extracellular recordings in urethane anaesthetised CRH[Cre] mice show that Barr[CRH] neurons provide a probabilistic drive that generates co-ordinated voids or non-voiding contractions depending on the phase of the micturition cycle. CRH itself provides negative feedback regulation of this process. These findings inform a new inferential model of autonomous micturition and emphasise the importance of the state of the spinal gating circuit in the generation of voiding.

## Introduction

The regulated production, storage and elimination of liquid waste as urine (micturition) plays a critical homeostatic role in maintaining the health of organisms. Like breathing, this involves precisely co-ordinated autonomic (parasympathetic and sympathetic) and somatic motor drives and has both voluntary and autonomous (involuntary) control mechanisms. The power of the autonomous drive is illustrated by the challenge faced by anyone *'caught short'* away from a socially acceptable location for urination. Disorders of autonomous micturition (resulting in involuntary voiding) are seen in over-active bladder syndrome, enuresis and following frontal lobe lesions (*Drake et al., 2010*; *Banakhar et al., 2012*; *Nevéus, 2017*). Barrington's nucleus, also known as the pontine micturition centre, is a key site for the control of urination (*Barrington, 1925*). The prevailing concept of the neural control of micturition is that afferent information from the bladder is conveyed via the spinal cord to the brainstem and periaqueductal gray (PAG) in the midbrain where it is integrated with information from higher centres such as hypothalamus and cortex (*Blok and Holstege, 1997*; *Shefchyk, 2001*; *Drake et al., 2010*; *de Groat and Wickens, 2013*). The synaptic drive from these

centres is relayed to Barrington's nucleus which appears to be a key command point for micturition (*Valentino et al., 1994*; *Hou et al., 2016*; *Verstegen et al., 2019*).

When the bladder is full, a threshold is reached and a neural command to void is relayed from Barrington's nucleus to the lumbosacral parasympathetic neurons and urethral sphincter motoneurons. Lesions of Barrington's nucleus (*Barrington, 1925*) or acute transection of the pons abolishes micturition (*De Groat, 1975*; *Sadananda et al., 2011*). In contrast, supra-collicular decerebration or transection of PAG does not stop micturition in cats, rats or mice (*Takasaki et al., 2010*; *Sadananda et al., 2011*; *Ito et al., 2018*). Similarly, co-ordinated voiding is seen under anaesthesia (*Pavcovich and Valentino, 1995*; *Hou et al., 2016*; *Ito et al., 2017*; *Keller et al., 2018*; *Verstegen et al., 2019*) when the contextual element of volitional voiding is removed. This constitutes autonomous micturition. Electrical or chemical stimulation of Barrington's nucleus induces bladder contraction (*Holstege et al., 1986*; *Noto et al., 1989*; *Mallory et al., 1991*; *Sasaki and Sato, 2013*). Functional imaging studies in humans (*Blok et al., 1997*; *Nour, 2000*) and rats *Tai et al., 2009* found activity in the dorsal pons during voiding. Thus, Barrington's nucleus is pivotal in the voiding reflex and is part of the minimal spino-bulbospinal circuit that generates autonomous voids and is believed to be the pre-parasympathetic control centre.

One of the largest populations of Barrington's nucleus neurons expresses corticotropin releasing hormone (CRH) in humans (*Ruggiero et al., 1999*) and rodents (*Vincent and Satoh, 1984*; *Valentino et al., 1995*; *Valentino et al., 2011*; *Verstegen et al., 2017*) and their axons terminate in the vicinity of the sacral parasympathetic neurons (*Valentino et al., 2011*; *Hou et al., 2016*; *Verstegen et al., 2017*). The role of these CRH-positive neurons in Barrington's nucleus (Barr$^{CRH}$) has recently been explored using CRH$^{CRE}$ mice to enable specific opto- and chemo-genetic manipulation of their activity (*Hou et al., 2016*; *Keller et al., 2018*). These studies indicated that Barr$^{CRH}$ neurons were glutamatergic, their activation caused bladder contraction (*Hou et al., 2016*; *Keller et al., 2018*) and increasing their excitability increased the probability of micturition (*Verstegen et al., 2019*). A second smaller subgroup of oestrogen receptor type-1 positive neurons in Barrington's nucleus (Barr$^{ESR1}$) have been shown to be important for control of the urethral sphincter in voluntary scent marking with urine (*Keller et al., 2018*). Further a group of layer-5 pyramidal neurons in the primary motor cortex plays a role in the descending control of voluntary urination via their projections to Barrington's nucleus (*Yao et al., 2018*). A common feature of many of these functional studies is that they have focussed on volitional voiding behaviours (*Hou et al., 2016*; *Keller et al., 2018*), such as scent marking of males in the presence of females which depends on descending inputs to the brainstem to trigger the voiding behaviour. This has led to different hypotheses about the role of the Barr$^{CRH}$ neurons in mediating voids in conscious mice with several studies concluding that they play a supporting rather than a primary role in generating voids (*Hou et al., 2016*; *Keller et al., 2018*).

Single-unit recordings of micturition-related neurons in the vicinity of Barrington's nucleus in rats and cats showed multiple different patterns of activity with either increased or decreased firing during bladder contractions (*de Groat et al., 1998*; *Sugaya et al., 2003*; *Tanaka et al., 2003*; *Sasaki, 2005b*; *Sasaki, 2005a*). These results were thought to reflect the neural heterogeneity within Barrington's nucleus and/or be due to the complex neural circuits in nearby brainstem sites involved the regulation of other pelvic visceral functions. More recent microwire recordings of the dorsal pons in rats reported neurons in the vicinity of Barrington's nucleus that have more homogeneous firing patterns, characterised by tonic activity with phasic bursts that were temporally associated with the voiding phase of the micturition cycle (*Manohar et al., 2017*) but they also show bursts of activity between voids that were not associated with increases in bladder pressure. A common technical limitation of these pontine neural recordings is the difficulty of identifying specific cell populations during or after recordings. This has been addressed for populations of Barrington's neurons through fibre-photometry of genetically encoded calcium indicators in mice (*Hou et al., 2016*; *Keller et al., 2018*; *Yao et al., 2018*) to show that Barr$^{CRH}$ and Barr$^{ESR1}$ neuronal activity increases around the time of voiding/scent marking respectively. However, the limited temporal and spatial resolution of the indicator and technique limits the ability to address whether this activity drives or follows micturition behaviour and the associated increase in bladder pressure. Therefore, the exact role of the Barr$^{CRH}$ neurons in micturition and specifically in autonomous micturition remains unclear – although it has recently been suggested they may play a more prominent (but relatively weak) role in promoting voids in anaesthetised mice (*Verstegen et al., 2019*). It is presumed that they act as a

central control centre generating a pre-parasympathetic drive to the bladder but it is not known whether they are sufficient on their own to generate a co-ordinated void through their actions on spinal circuits.

Here, we study the role of BarrCRH neurons in the autonomous micturition cycle in anaesthetised mice using opto- and chemo-genetic interventions as well as recordings of the firing activity of identified BarrCRH neurons in vivo. This has informed the development of a model indicating that these BarrCRH neurons provide a probabilistic signal to spinal circuits that is gated to trigger either non-voiding bladder contractions, which enable inferences to be made about the degree of bladder fullness, or voiding if a threshold level of pressure has been reached.

## Results

### BarrCRH neurons modulate micturition

Recent studies have drawn apparently contrasting conclusions about the role and importance of BarrCRH neurons in the regulation of volitional voiding (*Hou et al., 2016*; *Keller et al., 2018*). To further define their role, the light-activated cation channel ChR2 (channelrhodopsin-2) was selectively expressed in BarrCRH neurons of CRHCRE mice (*Taniguchi et al., 2011*) using a Cre-dependent adeno-associated viral vector (AAV-EF1α-DIO-ChR2-mcherry) (*Figure 1A–C*, *Figure 1—figure supplement 1*). Saline infusion to the bladder in urethane anaesthetised mice produced a regular cycle of autonomous micturition (voids at ~5 min intervals, average infusion rate $23 \pm 4$ µl/min). Tonic unilateral activation of BarrCRH neurons (5–20 Hz x 20 ms, 465 nm light pulses, applied for three completed voiding cycles) produced an increase of micturition frequency (to $153.2 \pm 15.6\%$ of basal after 10 Hz stimulation, *Figure 1D*) manifesting as a significant shortening in inter-void interval associated with a reduction of the threshold pressure for voiding ($84.0 \pm 4.7\%$ at 10 Hz, *Figure 1E*). Similar illumination in control mice (CRHCRE mice injected with AAV-DIO-hm4Di-mCherry instead of ChR2) had no effect on voiding frequency.

Chemogenetic-inhibition of BarrCRH neurons was achieved by expressing an inhibitory DREADD (*Figure 2A*, bilateral injection of AAV-DIO-hM4Di to CRHCRE mice). Administration of Clozapine N-oxide (CNO, 5 mg/kg i.p) produced a prolonged reduction in the frequency of voids seen during a continuous infusion protocol (reduced to $66.8 \pm 6.3\%$ of control at 20 mins, *Figure 2B*). This was associated with increases of threshold pressure, fill volume and micturition pressure (*Figure 2C*).

The chemogenetic inhibition of BarrCRH during this continuous filling protocol caused the bladder to become increasingly distended leading to incomplete voids. To control for this effect a 'fill and hold' protocol was employed to fill the bladder to a maximum pressure of 15 mmHg (close to the threshold pressure for voiding). This volume was held for up to 10 min until either the mouse voided spontaneously, or the bladder was manually emptied, and the filling cycle restarted (*Figure 2D*). Using this protocol, chemogenetic inhibition caused a prolongation of the latency to void (from $207 \pm 17$ s at baseline to $738 \pm 60$ s at 20mins after CNO). This effectively produced a period of urinary retention, that persisted for 2 hr (*Figure 2E*).

These findings indicate that BarrCRH neurons have a potent ability to modulate the autonomous micturition cycle and that their basal level of activity is of functional importance.

### BarrCRH neurons do not simply act as high-fidelity controllers of bladder pressure

To assess whether BarrCRH neurons act as a tightly-coupled, pre-motor drive to bladder parasympathetic neurons (*Fowler et al., 2008*; *de Groat and Wickens, 2013*) bladder pressure was recorded while parametrically opto-activating Barrington's nucleus unilaterally ($9.5 \pm 0.3$ mW, 465 nm). In initial experiments, unilateral opto-activation of BarrCRH (20 ms x 20 Hz for 5 s) evoked non-voiding contractions of the bladder (eNVC, *Figure 3A and B*, with the bladder filled to half of its threshold capacity, *n = 7* mice). These eNVC were similar to the transient bladder contractions triggered by optoactivation of BarrCRH neurons previously noted by *Hou et al. (2016)*. Varying stimulus frequencies and pulse durations produced modestly graded changes in eNVC with a 20 ms x 20 Hz protocol producing near maximal responses ($3.9 \pm 0.8$ mmHg, *Figure 3C and D*). The eNVC had a consistent latency to onset of $1.3 \pm 0.1$ s and a time to peak of $6.0 \pm 0.3$ s following stimulus onset and an average duration of $8.2 \pm 0.6$ s. With each of the stimulus parameters there were 'failures' where there

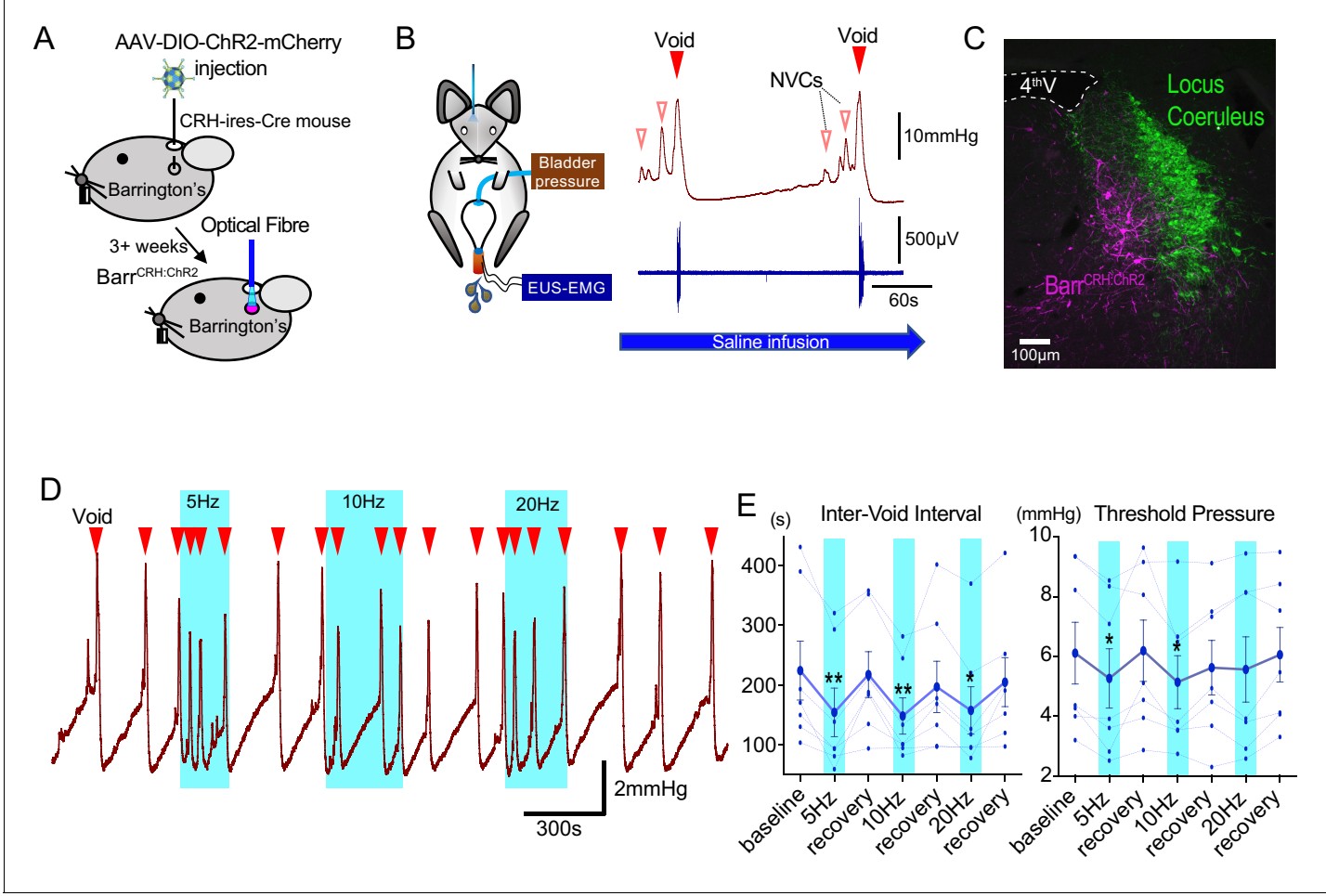

**Figure 1.** Optoactivation of Barr[CRH] neurons shortens the micturition cycle. (**A**) AAV-DIO-ChR2-mCherry injection to dorsal pons in CRH[Cre] mice followed after 3 weeks by opto-activation by light (465 nm) from an optical fibre. (**B**) In vivo recording in urethane anaesthetized mice with optical fibre positioned above Barrington's nucleus in the pons with bladder pressure and external urethral sphincter activity monitoring during the micturition cycle (with continuous bladder filling). (**C**) Post-hoc histology after stereotaxic injection of AAV-DIO-ChR2-mCherry demonstrating transduction of Barr[CRH] neurons in sections of dorsal pons. Immunohistochemistry for mCherry (Magenta) shows transduced Barr neurons and Tyrosine hydroxylase (Green) marks the adjacent Locus Coeruleus. (**D**) Periods of maintained unilateral opto-activation of Barr[CRH:ChR2] increased the frequency of micturition (light pulsed at 5, 10 and 20 Hz x 20 ms for three micturition cycles). (**E**) Continuous opto-activation (5, 10 and 20 Hz) reversibly shortened the inter-void interval (n = 7 mice). Similarly, the threshold for voiding was reversibly reduced with 5 and 10 Hz stimulation (Friedman test with Dunn's multiple comparisons to prior unstimulated state *p<0.05, **p<0.01, *Figure 1—source data 1*). NVC – Non-voiding contraction. Cystometrogram measures shown on *Figure 1—figure supplement 2*.

The online version of this article includes the following source data and figure supplement(s) for figure 1:

**Source data 1.** Data for 'Optoactivation of BarrCRH neurons shortens the micturition cycle'.
**Figure supplement 1.** Histological verification of vector expression in Barrington's nucleus.
**Figure supplement 2.** Cystometrogram.

was no detectable bladder response (*Figure 3 and E*). The probability of eNVC increased with stimulation frequency (71.4 ± 7.6% at 2.5 Hz and 97.1 ± 2.5% at 20 Hz) with 20 Hz being the most reliable. Single light pulses of longer duration (1–3 s) were also able to reliably generate eNVC. In contrast, illumination in control mice (CRH[CRE] mice injected with AAV-DIO-hm4Di-mCherry instead of ChR2) had no effect on the bladder pressure (101.4 ± 0.6% compared to the pressure immediately before optoactivation at 20 Hz x 20 ms, n = 3).

Previous anatomical studies with retrograde tracing using pseudorabies virus have suggested that there is a route for communication between afferent neurons innervating the distal colon and Barrington's nucleus and further that Barrington's neurons were activated by distention of the distal

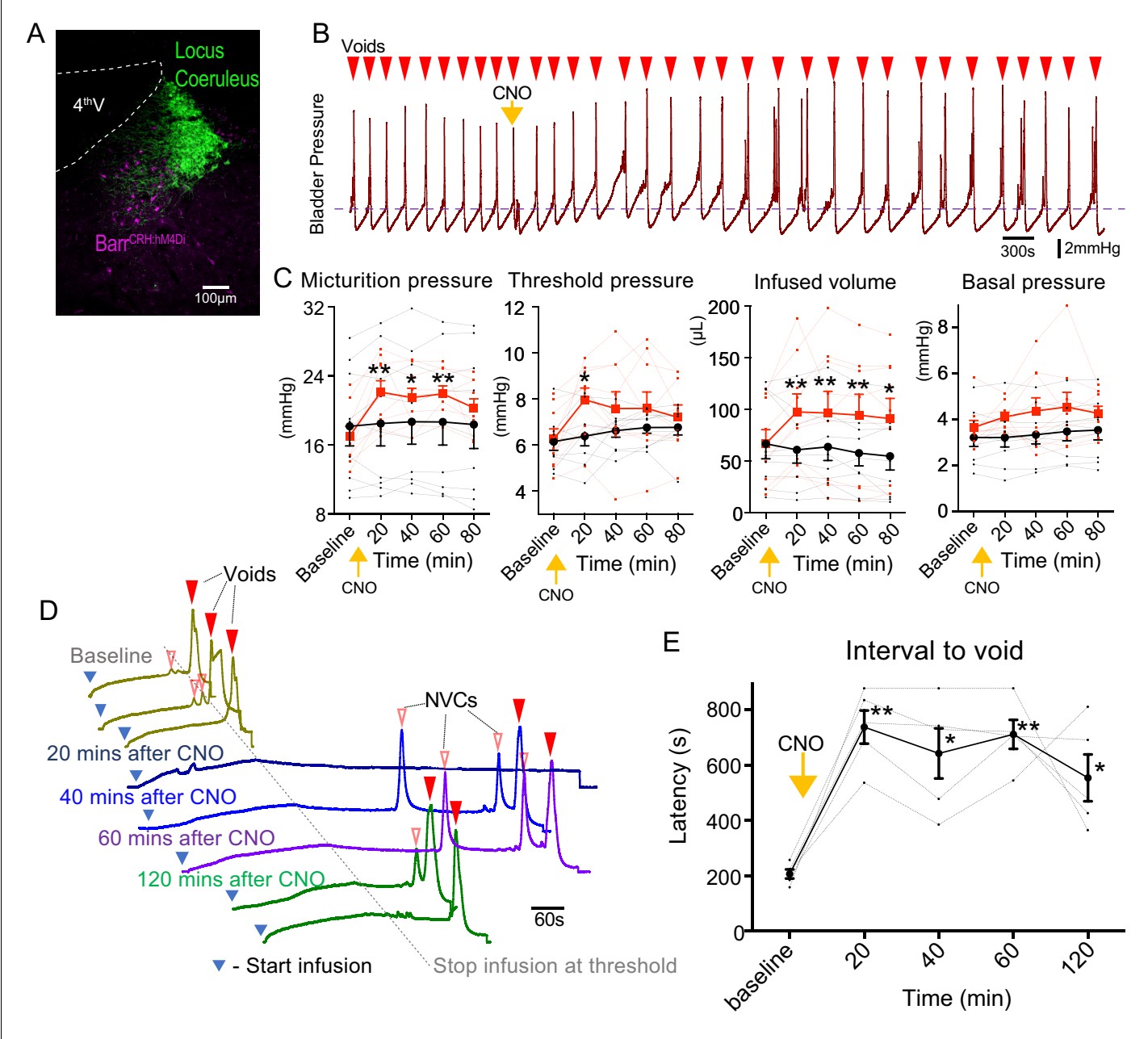

**Figure 2.** Chemogenetic inhibition of Barr[CRH] neurons prolongs the micturition cycle. (A) Transduction of Barr[CRH] with inhibitory DREADD using AAV-DIO-hM4Di-mCherry demonstrated with immunocytochemistry for mCherry (magenta) and Tyrosine hydroxylase (green) to mark the Locus Coeruleus. (B) Administration of the DREADD ligand CNO (5 mg/kg, i.p) slowed the frequency of micturition seen with continuous saline infusion to the bladder. (C) The chemogenetic inhibition of Barr[CRH:hM4Di] neurons caused an increase in the voiding threshold (129.7 ± 8.9%), volume infused before void (161.9 ± 16.9%) and micturition pressure (131.7 ± 6.9%) compared to baseline (RM-ANOVA with Holm-Sidak's post hoc, *p<0.05, **p<0.01) unlike control mice (Barr[CRH:ChR2], $n = 9$ per group) where CNO was without significant effect. In each case this CNO effect peaked around 20 min after administration and reversed slowly. (D) Using an intermittent bladder infusion protocol (to a maximum bladder pressure of 15 mmHg) CNO administration inhibited voiding with (E) a large increase in the latency to void (time after start of infusion) – equivalent to urinary retention ($n = 5$) (RM-ANOVA with Holm-Sidak's post hoc, *p<0.05, **p<0.01). Source data in *Figure 2—source data 1*.

The online version of this article includes the following source data for figure 2:

**Source data 1.** Data for 'Chemogenetic inhibition of BarrCRH neurons prolongs the micturition cycle'.

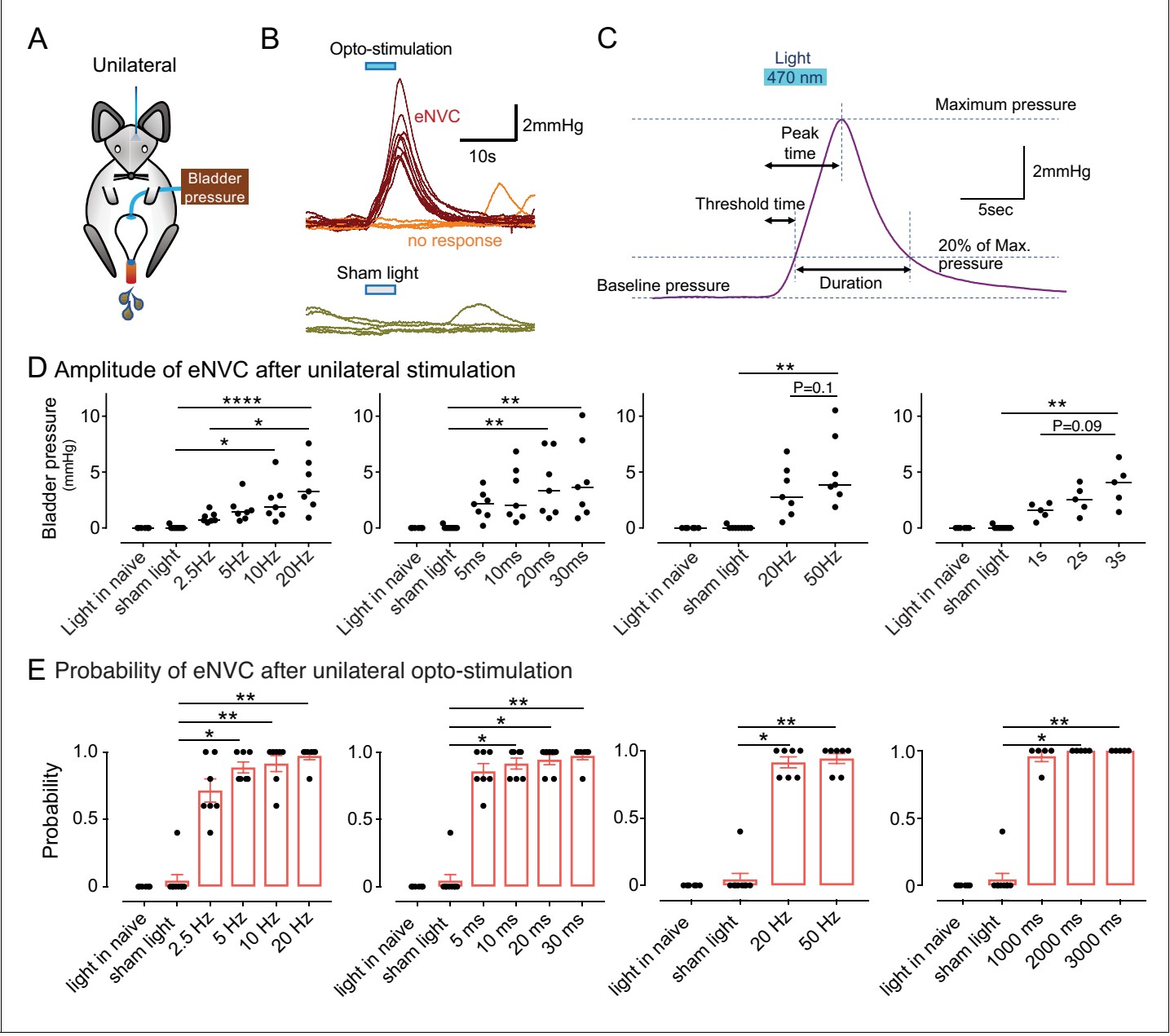

**Figure 3.** Phasic optoactivation of Barr[CRH] evokes bladder contractions. (**A**) Bladder pressure recordings with unilateral opto-activation (**B**) Phasic opto-activation (20 ms x 20 Hz, 5 s) of Barr[CRH] neurons evoked non-voiding contractions (eNVCs, with the bladder ~half full, static). These eNVCs had a stereotyped shape and a relatively constant latency. In addition, there were 'failures' where no response was evoked by an identical stimulus. (**C**) Parameters of Barr[CRH] evoked non-voiding contractions. Threshold calculated at 20% of the amplitude, duration was measured at the threshold pressure. The latency was taken as the time from start of stimulation for the pressure to reach threshold. (**D**) The amplitude of eNVC increased with stimulation frequency (pulse length 20 ms for 5 s) and pulse duration (at 20 Hz for 5 s) (n = 7 mice). Higher frequencies of stimulation (50 Hz x10ms) did not substantially increase eNVC amplitude. Single longer light pulses (1–3 s) could also generate graded eNVCs. (**E**) The probability of generating an eNVC increased with stimulation frequency (pulse length 20 ms for 5 s) and pulse duration (frequency 20 Hz for 5 s). Longer light pulses (1–3 s) also reliably generated eNVCs. (RM-ANOVA with Dunnett's post hoc or Friedman's test, *p<0.05, **p<0.01, ****p<0.0001). Source data in *Figure 3—source data 1*.

The online version of this article includes the following source data and figure supplement(s) for figure 3:

**Source data 1.** Data for 'Phasic optoactivation of BarrCRH evokes bladder contractions'.
**Figure supplement 1.** Barr[CRH] optoactivation does not cause contraction of the distal colon.

colon (*Rouzade-Dominguez et al., 2003*). This has led to the suggestion that Barrington's nucleus may control the lower gastrointestinal tract as well as the lower urinary tract. Indeed, defaecation was suggested to occur on occasion (but not quantitated) following opto-activation of Barr[CRH] neurons in mice implying a role in motor control (*Hou et al., 2016*). To investigate a possible relationship between Barr[CRH] and activity in the distal colon, a balloon catheter was inserted to monitor pressure in pilot experiments (n = 2 mice). Distal colonic pressure was not synchronised with bladder pressure during the normal micturition cycle (*Figure 3—figure supplement 1B*). Furthermore, optogenetic activation of Barr[CRH] neurons did not alter distal colonic pressure (*Figure 3—figure supplement 1C and D*), despite the generation of bladder eNVC. This preliminary evidence suggests that this Barr[CRH] population of Barrington's neurons is not involved in motor control of the colon.

It was postulated that recruitment of a larger population of Barr[CRH] neurons in synchrony might be more effective in triggering larger or more reliable bladder contractions. This was tested with bilateral expression of ChR2 and a dual-fibre optical cannula allowing independent activation of one, the other or both Barrington's nuclei (*Figure 4A–C*). Bilateral activation of Barr[CRH] produced larger eNVC (7.1 ± 2.0 mmHg at 20 ms and 20 Hz, n = 7 mice) than optoactivation of either side alone (2.9 ± 0.5 and 3.1 ± 0.8 mmHg for right or left side, respectively), particularly at higher frequencies of stimulation (*Figure 4D and E*). The effect of bilateral stimulation on eNVC amplitude was additive rather than synergistic. The probability of generating eNVC was increased by bilateral stimulation (evident at lower stimulus frequencies that is increased by 154 ± 18% for bilateral *vs* right alone or by 158 ± 17% for bilateral *vs* left alone at 2.5 Hz, *Figure 4E*). It was notable that, with the bladder filled to half of its threshold capacity, bilateral Barr[CRH] stimulation never triggered voids.

These findings support the proposal that Barr[CRH] neurons can selectively generate bladder contractions and indicate that this is a probabilistic process, with failures, rather than being a simple high-fidelity pre-motor drive to the bladder.

## Bladder pressure responses to Barr[CRH] drive augments with progress through the micturition cycle

This raised the question of whether the stage of the micturition cycle influences the bladder pressure response to Barr[CRH] opto-activation as the cycle phase may modulate Barr[CRH] neuronal excitability. During continuous bladder filling, it was noted that the amplitude of Barr[CRH] eNVC increased progressively through the micturition cycle (increase of 17.0 ± 3.9 fold, comparing eNVC obtained during the 2[nd] versus 5[th] quintile of micturition cycle, *Figure 5A–D*). This phenomenon was also observable, albeit not quantitated or commented upon, in the recordings of *Hou et al. (2016)*, see *Figure 5B*). Similarly, the probability of obtaining a bladder contraction with optoactivation also increased with progressive filling, with most 'failures' being seen when the bladder was <40% filled (*Figure 5E*). The same phase dependence of eNVC was also apparent with bilateral stimulation of Barr[CRH] (*Figure 5—figure supplement 1A and B*).

The phase-dependence of eNVC amplitude may, in part, be a consequence of bladder distension, leading to raised passive detrusor tension and an increase of length-dependent contractions. To test this proposition, the effect of pelvic nerve stimulation was assessed in the pithed decerebrate, arterially-perfused mouse preparation (*Ito et al., 2018*; *Ito et al., 2019*). The amplitude of bladder contractions induced by pelvic nerve stimulation (4–20 Hz, 10V, 3 s) increased with bladder filling (*Figure 5—figure supplement 2*) with a doubling (2.2 ± 0.34 fold at 20 Hz) of the pressure generated between empty bladder and 70 μl fill (close to voiding threshold in an intact mouse). However, this amplitude increase plateaued at a volume of ~50 μl – and showed a much less steep relationship than that observed for Barr[CRH] eNVC in vivo which increased by 17-fold over the same range of bladder distension. Additionally, this relationship did not account for the observed probabilistic nature of eNVC, as failures were never observed with pelvic nerve stimulation.

## Barr[CRH] stimulation can conditionally trigger complete voids

Although tonic stimulation of Barr[CRH] increased voiding frequency, it was not possible to trigger full voiding contractions with phasic Barr[CRH] stimulation with the bladder up to 50% filled, even with bilateral stimulation. However, by applying stimuli systematically at points through the micturition cycle it was possible to trigger fully co-ordinated voids by activating Barr[CRH] neurons later in the cycle (>50% filled, *Figure 5B–D*). The pattern and amplitude of the evoked bladder contraction was

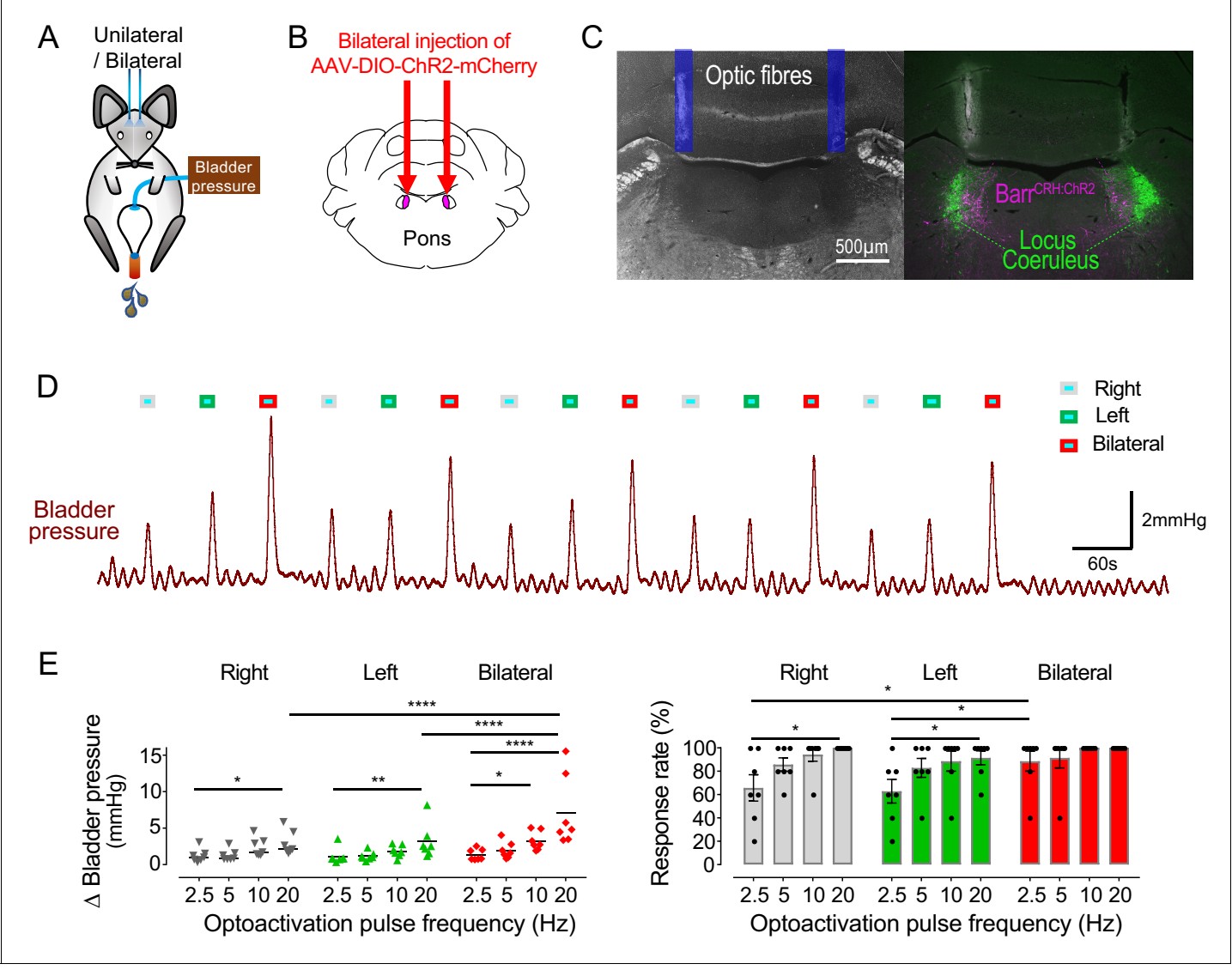

**Figure 4.** Bilateral phasic optoactivation of Barr[CRH] evokes larger non-voiding contractions than unilateral optoactivation. (**A**) Bladder pressure recordings with unilateral or bilateral opto-activation following (**B**) bilateral injection of AAV-DIO-ChR2-mCherry. (**C**) Confirmation of bilateral Barr[CRH:ChR2] transduction and optic fibre targeting (immuno for mCherry – magenta and TH - green). (**D**) Bladder pressure response showing the effect of unilateral or bilateral phasic opto-activation (20 ms x 20 Hz, 5 s) of Barr[CRH] neurons evoked non-voiding contractions (eNVCs, with the bladder ~half full, static). A comparison of the effects of unilateral versus bilateral stimulation (**E**) showed that bilateral stimulation evoked larger events and with an increased reliability than either side alone (each point represents data from a single mouse, $n = 7$ mice, RM-ANOVA with Dunnett's post hoc or Friedman's test, *$p<0.05$, **$p<0.01$, ****$p<0.0001$). Source data in *Figure 4—source data 1*.

The online version of this article includes the following source data for figure 4:

**Source data 1.** Data for 'Bilateral phasic optoactivation of BarrCRH evokes larger non-voiding contractions than unilateral optoactivation'.

similar to that seen with spontaneous voids and they occurred at a similar latency to eNVC. In addition, voiding was complete and the empty bladder relaxed to the basal pressure level after each void.

The mouse external urethral sphincter (EUS) shows bursting activity during spontaneous voids which facilitates urine expulsion (*Ito et al., 2018*; *Keller et al., 2018*). Injections of pseudorabies virus into either the bladder or EUS has shown labelling in the vicinity of Barrington's nucleus, suggesting it is part of the EUS control circuit (*Nadelhaft et al., 1992*; *Nadelhaft and Vera, 1996*; *Marson, 1997*). However, recent evidence suggests that it is the Barr[ESR-1], rather than Barr[CRH], neurons which project to local circuit interneurons in L4-5 that may regulate EUS motoneurons (*Keller et al.,*

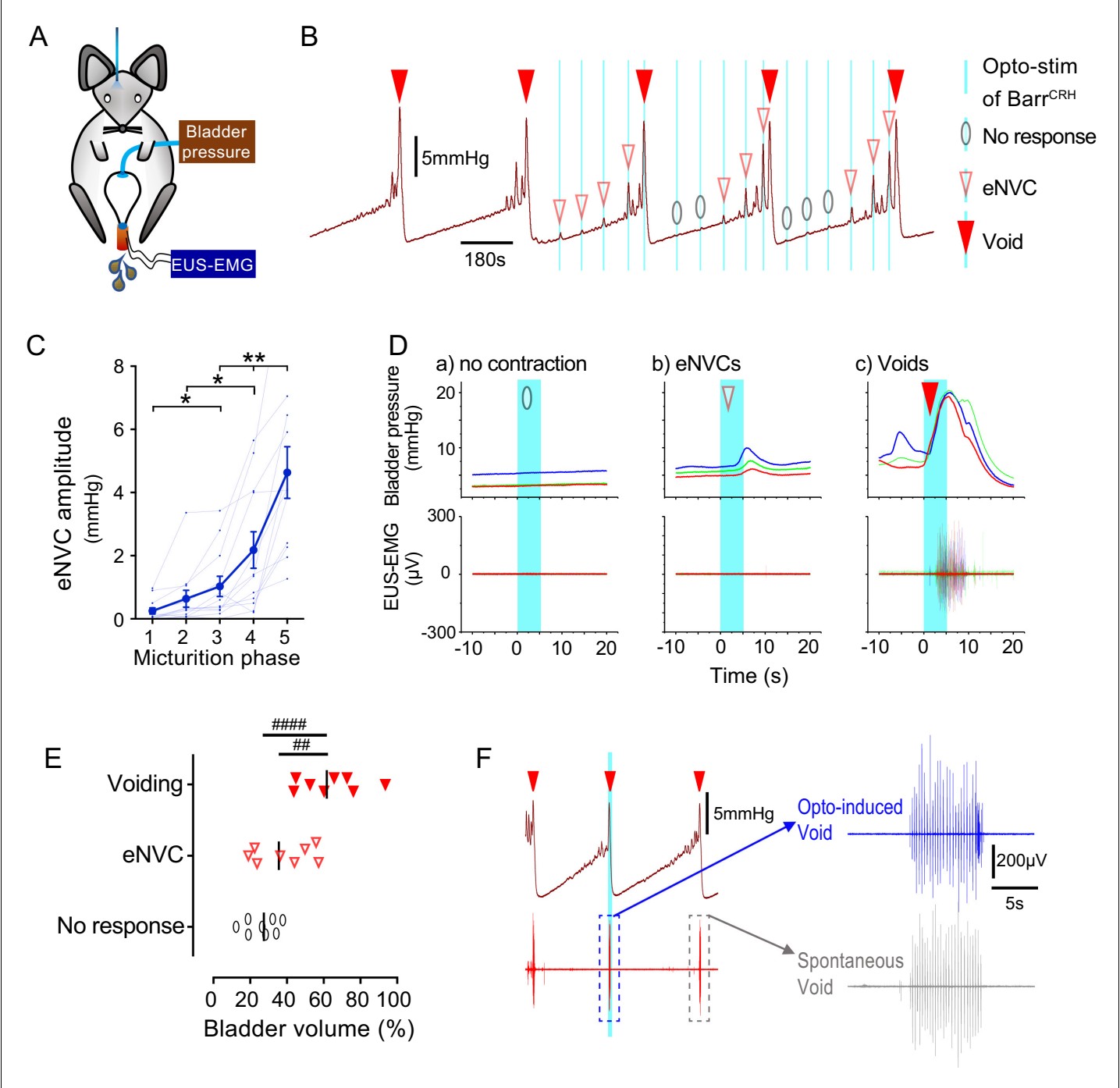

**Figure 5.** Dynamics of Barr[CRH] evoked events through the micturition cycle. (**A**) Experimental set-up with unilateral opto-activation of Barr[CRH] neurons. (**B**) Continuous infusion cystometry with episodic opto-activation (20 Hz x 20 ms for 5 s) applied at different phases of the micturition cycle generates eNVC of incrementing amplitude as the cycle progresses. (**C**) There is a substantial increase in the amplitude of the eNVC as the micturition cycle progresses (17.0 ± 3.9 fold comparing eNVC from the 2nd and 5th quintiles of the cycle) (RM one-way ANOVA followed by Dunnet's test, *-P < 0.05, **-P < 0.01). (**D**) Overlaid bladder pressure responses to the same optogenetic stimulus applied (x3) at different phases of the cycle can trigger either no response or eNVCs or full voiding contractions that show a stereotyped morphology and latency. (**E**) Analysis of the stage of the voiding cycle where each type of response was triggered showed that voiding contractions were significantly more likely to be evoked later in the voiding cycle (each symbol represents the average position of such events in each mouse, n = 8) (RM one-way ANOVA with Tukey's test, ##-P < 0.01, ####-P < 0.0001). (**F**) The bursting pattern of EUS activity was similar with both Barr[CRH]-evoked and spontaneous voids. Source data in *Figure 5—source data 1*. The online version of this article includes the following source data and figure supplement(s) for figure 5:

**Source data 1.** Data for 'Dynamics of BarrCRH evoked events through the micturition cycle'.

*Figure 5 continued on next page*

*2018*) analogous to the lumbar spinal coordinating centre (LSCC; *Chang et al., 2007*). Therefore, recordings were made from the EUS to investigate the relationship of the voiding-associated bursting to Barr^CRH activation. The Barr^CRH eNVC (irrespective of their magnitude) were never associated with EUS activity (*Figure 5B and D*).

However, when Barr^CRH activation evoked a voiding contraction then bursting EUS activity was always found (*Figure 5D and F*). These Barr^CRH induced voids had EUS activity that was indistinguishable from spontaneous voids in terms of burst duration (spontaneous $4.4 \pm 0.9$ vs opto-induced $4.4 \pm 1.0$ s, n = 7, paired t-test, ns) and frequency (spontaneous $22.7 \pm 2.7$ vs opto-induced $21.5 \pm 3.0$ Hz, n = 7, paired t-test, ns)). These results indicate that the Barr^CRH neurons can trigger voids that are in all aspects similar to those seen spontaneously but that can be triggered to occur earlier in the normal micturition cycle.

## Spinal drive from Barr^CRH neurons is sufficient to generate eNVC and voids

The axons from Barr^CRH were noted to provide a specific innervation of the sacral parasympathetic neurons but not to the ventral horn at the level of Onuf's nucleus (*Figure 6A*, *Figure 6—figure supplement 1*). To investigate whether optogenetic stimulation of spinal axons of Barr^CRH is sufficient to directly generate eNVC, bladder pressure was recorded while light was applied from an optic fibre located above the spinal cord. Optogenetic stimuli (either 20 ms x 20 Hz for 5 s or single 1 s pulse) applied to the spinal cord reliably induced bladder contractions (*Figure 6B–D*, p=0.025, bladder half filled). These eNVCs tended to occur with a shorter latency than those evoked directly from pontine stimulation ($1.0 \pm 0.2$ s vs $1.26 \pm 0.1$ s, *n = 5*). Similarly, during continuous bladder filling, spinal activation could trigger full voids (*Figure 6E*). These data support the principle that the Barr^CRH neurons can evoke both voiding and eNVC through their spinal projections.

## Spinal CRH inhibits the bladder response to Barr^CRH activation

It has been proposed that CRH released from Barrington's neurons at a spinal level augments bladder pressure responses (*Klausner and Steers, 2004*; *Klausner et al., 2005*) although others have reported the opposite action (*Pavcovich and Valentino, 1995*; *Kiddoo et al., 2006*; *Wood et al., 2013*) and genetic knock out of CRH expression in Barrington's neurons was without phenotype (*Verstegen et al., 2019*). If the release of CRH does increase during the micturition cycle, then this might be predicted to act as a positive feedforward mechanism to augment the parasympathetic and hence bladder pressure responses to Barr^CRH drive. To test this hypothesis, the effect of intrathecal Astressin (a broad-spectrum CRH antagonist, 5 μg in 5 μl) on Barr^CRH eNVC was assessed through the micturition cycle. Counter to the prediction, Astressin significantly and reversibly increased the amplitude of eNVC, an action that was more pronounced as the bladder filled ($333 \pm 75\%$, p=0.008 (*n = 7*), 20 mins after Astressin, *Figure 7*). Intrathecal Astressin also decreased the infused volume required to trigger a void (*Figure 7D*).

This indicates that CRH is providing a negative feedback signal to limit the extent of the spinal parasympathetic response to Barr^CRH neuronal activity (in agreement with *Pavcovich and Valentino, 1995*; *Kiddoo et al., 2006*; *Wood et al., 2013*). Therefore, increased release of CRH cannot account for the augmented responses to Barr^CRH activation with progression through the micturition cycle.

## Barr^CRH activity anticipates bladder pressure during the micturition cycle

Neural recordings from cats (*Sasaki, 2005a*) and rats (*Manohar et al., 2017*) indicates that some putative Barrington's neurons fire intermittently during the storage phase with an increase of firing that occurs around voiding, consistent with a role in mediating the drive to bladder parasympathetic

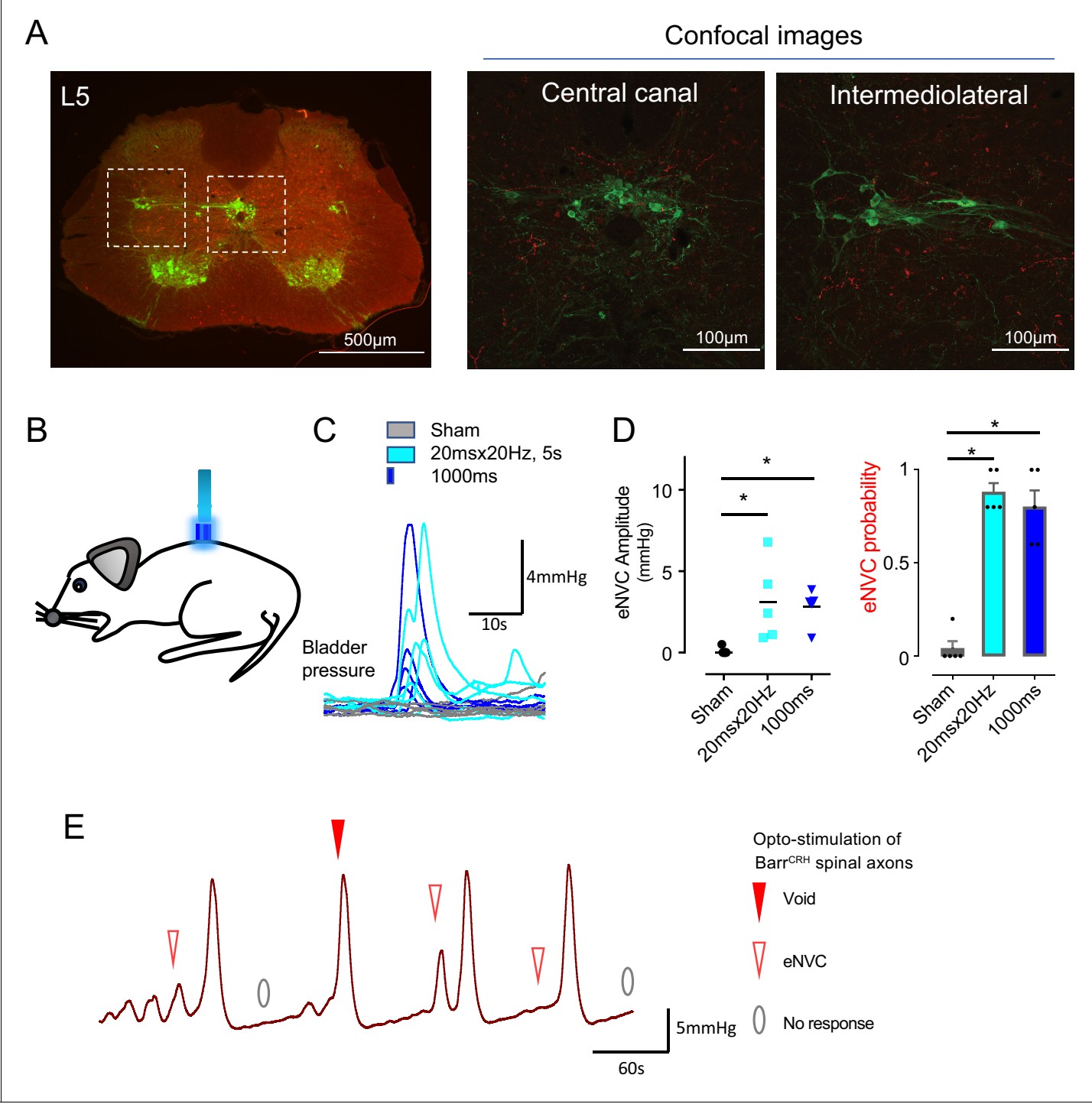

**Figure 6.** Spinal opto-activation of Barr[CRH] axons generates eNVC and voids. (**A**) Unilateral transduction of Barr[CRH] neurons with AAV-EF1α-DIO-ChR2-mCherry. Spinal L5 section had immunocytochemistry for mCherry (red) and Choline acetyltransferase (green) to label filled Barr[CRH] axons and somatic and autonomic motoneurons. The Barr[CRH] axons show a lateralised distribution targeting the territory of parasympathetic preganglionic neurons at L5 (**B**) The spinal cord was exposed at the vertebral level of T11-12 and illuminated from an optic fibre placed above the cord. (**C**) Opto-activation (20 Hz x 20 ms for 5 s or single 1 s pulse) generated eNVCs (Related samples Friedman's test by ranks). (**D**) There was no difference in the eNVC in terms of amplitude or reliability between the two opto-stimulus patterns (n = 5 mice). (**E**) Opto-stimulation (20 Hz x 20 ms for 5 s) during continuous filling cystometry generated full voiding contractions as well as eNVCs. Source data in *Figure 6—source data 1*.

The online version of this article includes the following source data and figure supplement(s) for figure 6:

**Source data 1.** Data for 'Spinal opto-activation of BarrCRH axons generates eNVC and voids'.

*Figure 6 continued on next page*

*Figure 6 continued*

**Figure supplement 1.** Spinal projections of Barr[CRH] axons.

neurons. Recent fibre photometric recordings of Barr[CRH] neurons, using the genetically encoded calcium indicator GCaMP6, indicate that the activity of these neurons is 'in phase' with the micturition cycle (*Hou et al., 2016*; *Keller et al., 2018*). However, fibre photometry is unable to resolve the

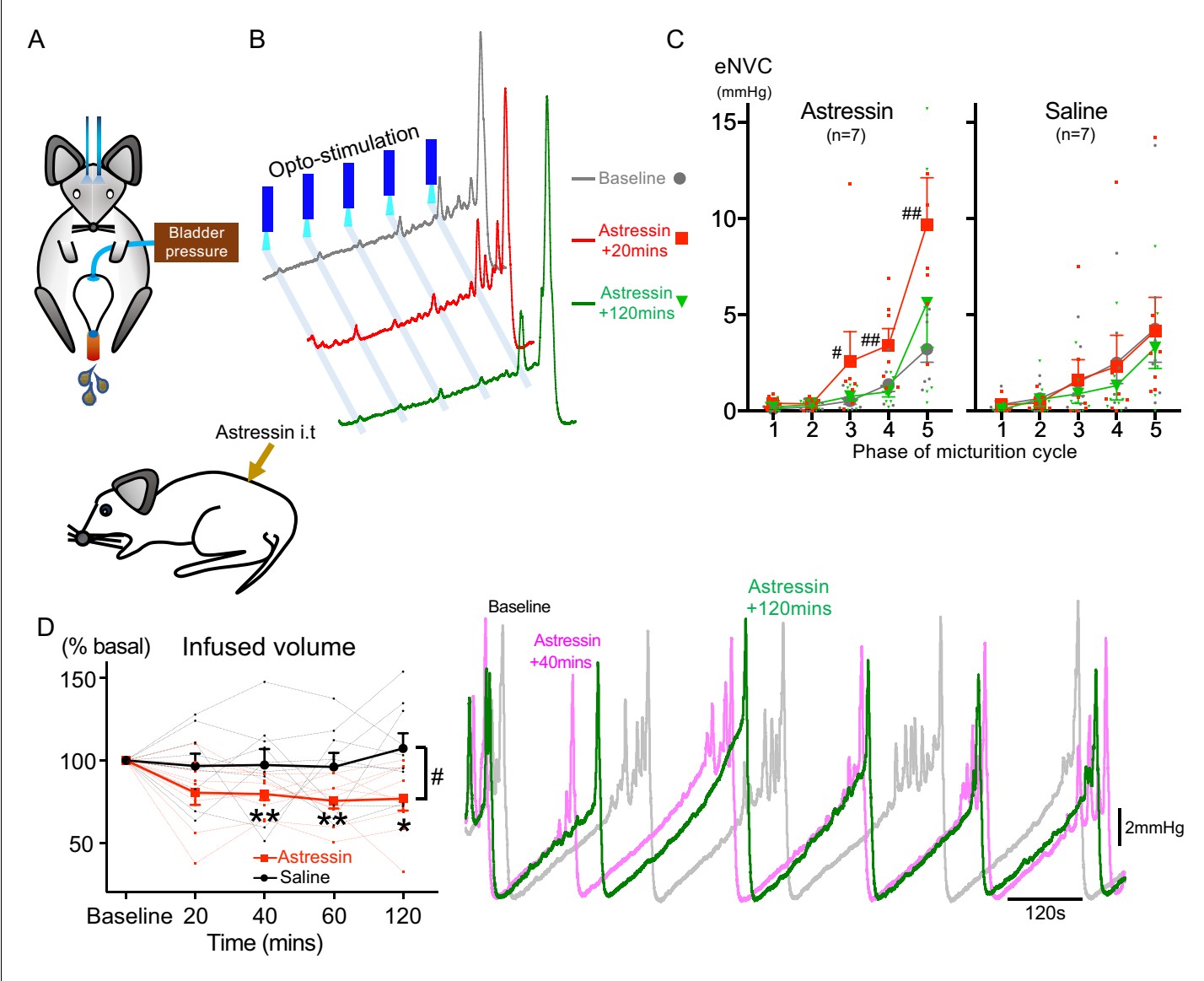

**Figure 7.** Spinal CRH inhibits the bladder response to Barr[CRH] neuronal optoactivation. (**A**) Assessment of the influence of intrathecal Astressin (CRH antagonist) on the bladder pressure response to bilateral optoactivation of Barr[CRH] neurons. (**B**) Intrathecal Astressin (5 μg) reversibly increased the amplitude of Barr[CRH] eNVC (*n = 7* mice). (**C**) Summary data for the action of intrathecal Astressin on eNVC (versus vehicle control) showing that the augmentation of amplitude was particularly marked towards the end of the micturition cycle (Related samples Friedman's test by ranks, #-P < 0.05, ##-P < 0.01). (**D**) Even without Barr[CRH] opto-stimulation Astressin reversibly increased the frequency of voiding compared both to baseline and an intrathecal vehicle control group (*n = 9*) (vs baseline with related samples Friedman's test by ranks and vs vehicle with Mann-Whitney U test, *-P < 0.05, **-P < 0.01). Source data in *Figure 7—source data 1*.

The online version of this article includes the following source data for figure 7:

**Source data 1.** Data for 'Spinal CRH inhibits the bladder response to BarrCRH neuronal optoactivation'.

action potential discharge patterns from Barr[CRH] neurons in vivo. As such it has not previously been possible to directly assess the functional relationship between Barr[CRH] firing and bladder pressure.

Neuronal activity was recorded in the vicinity of Barrington's nucleus using a 32-channel silicon probe to test whether changes in the excitability of Barr[CRH] neurons during the micturition cycle accounts for the observed variation in the evoked pressure responses of the bladder. An optic fibre was placed above Barrington's nucleus enabling optogenetic identification (*Figure 8A*). Recordings were made of cell activity during the normal micturition cycle (with simultaneous bladder pressure and EUS EMG activity) and in response to the application of light stimuli. A total of 113 individual neurons were identified by clustering from recordings made in the vicinity of Barrington's nucleus (*n = 3* mice, *Figure 8BD*). Definitive opto-identification of Barr[CRH] neurons (*n = 12*) was indicated by reliable short latency spike entrainment to light (20 ms pulses, *Figure 8C*) with time-locked, maintained firing in response to longer light pulses (≥1 s, *Figure 8C*).

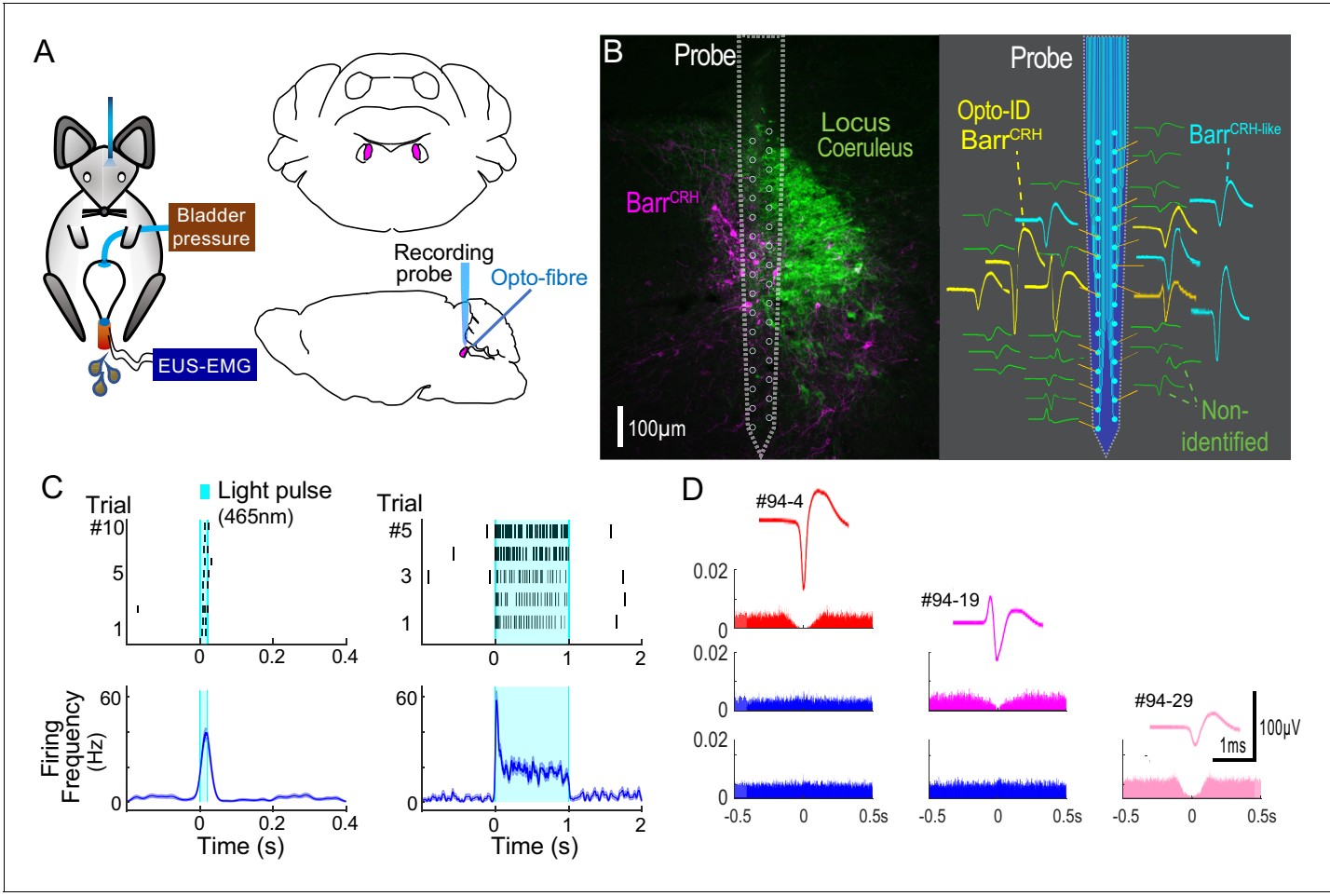

**Figure 8.** Multiunit recordings of identified Barr[CRH] neurons. (A) Schematic with unilateral stimulation and recording of Barrington's nucleus with simultaneous bladder and EUS monitoring. (B) Immunohistochemistry (mCherry - magenta and TH - green) confirming the position of the recording electrode (shown to scale and with its tip at the end of the histological track). The spike waveforms of individual units are shown schematically adjacent to their probe recording site. Note that the Barr[CRH] neurons (yellow) are clustered in sites located within Barrington's nucleus whereas the non-identified neurons (green) lie above and below the level of Barrington's nucleus. A third population of non-optoidentified neurons is shown in blue (labelled Barr[CRH-like]) whose firing pattern closely resembled the Barr[CRH] neurons (C) Barr[CRH] neurons were optoidentified by a short latency response to a brief light pulse (20 ms) data shown for a single representative unit top left. The population response of identified Barr[CRH] neurons shown below (*n = 12*, smoothed average firing rate curve generated by convolution of spikes with a Gaussian of SD 10 ms). The response to a 1 s light pulse is shown to the right with the same single unit and the population response from all Barr[CRH] neurons. Note that they showed an initial high frequency response that decayed to a plateau of ~20 Hz likely reflecting the kinetics of ChR2 currents. (D) Auto- and cross-correlations (1 ms bin size) of three opto-identified Barr[CRH] neurons with their average spike waveforms showing isolation and a degree of cross-correlation at short latency.

These BarrCRH neurons showed a characteristic pattern of activity during the micturition cycle with bursting at the time of voiding (*Figure 9A*, 20.5 ± 4.1 Hz peak firing frequency). A second population of neurons was recorded with a similar pattern of activity (but were not activated by light) that are henceforth termed BarrCRH-like (*n = 32*, *Figures 9B* and *10A,B*) in distinction to the remainder of non-identified neurons (*n = 69*). These BarrCRH-like neurons had a short-latency synchrony with the

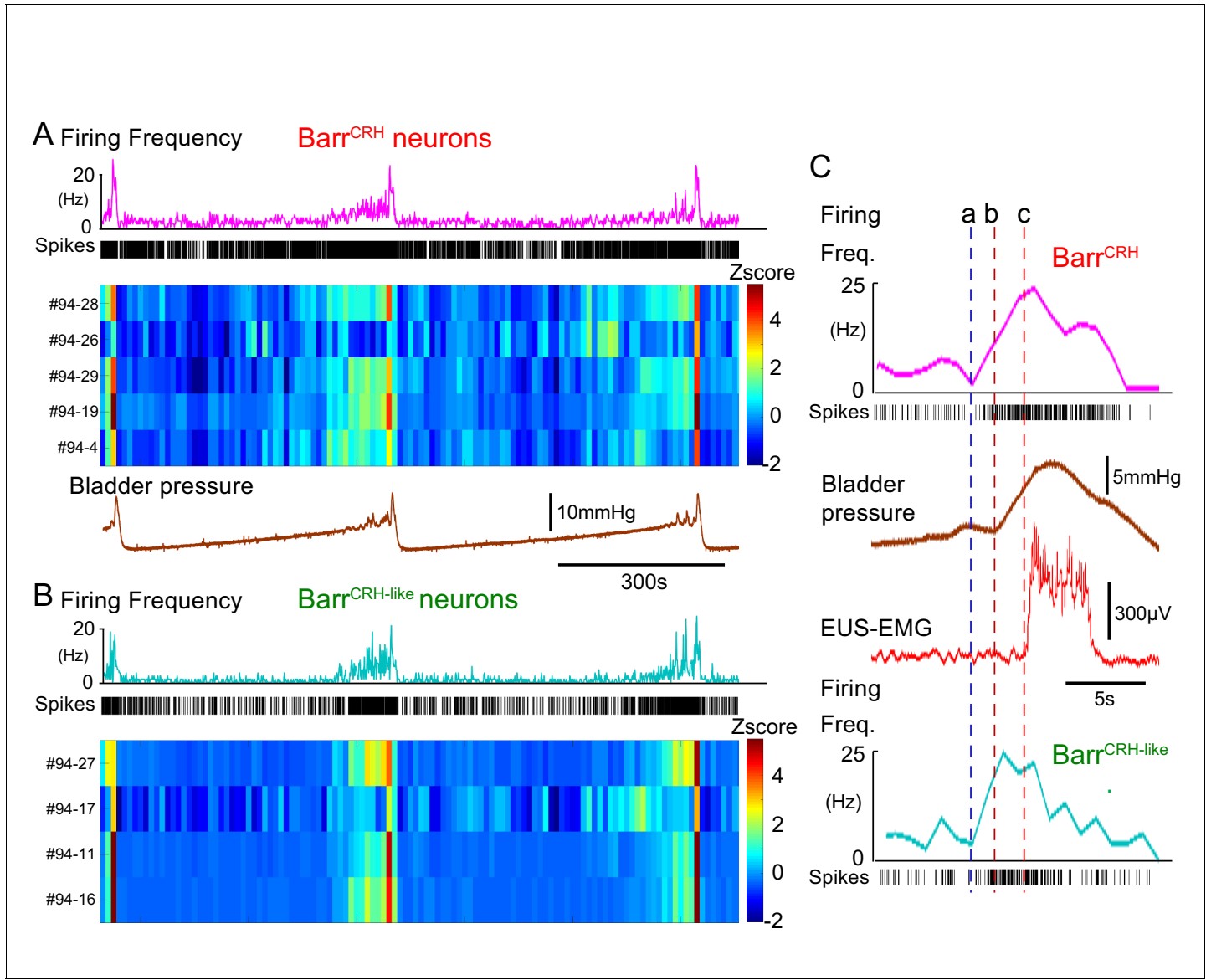

**Figure 9.** BarrCRH neuronal firing anticipates bladder pressure during the micturition cycle. (A) BarrCRH neurons showed a bursting pattern of discharge that aligned with bladder pressure. The z-scored responses of all BarrCRH neurons in this recording can be seen to have a similar pattern of activity (single representative firing rate plot shown above). (B) Within the same recording (and from adjacent probe sites) a further group of neurons was noted (n = 4) to exhibit a similar pattern of bursting discharge synchronized to the voiding cycle. These neurons were termed BarrCRH-like. Auto and cross-correlations of the BarrCRH-like and BarrCRH neurons (see *Figure 9—figure supplement 1*) showed them to have similar properties and evidence of a degree of short latency correlation to other BarrCRH-like neurons and also BarrCRH neurons. (C) The increase in firing activity (a) of both BarrCRH-like and BarrCRH neurons (same experiment), preceded and anticipated the change in bladder pressure (b) and occurred before the onset of voiding marked by the sudden increase in EUS-EMG (c).

The online version of this article includes the following figure supplement(s) for figure 9:

**Figure supplement 1.** *Auto- and cross-correlations of BarrCRH and BarrCRH-like neurons* BarrCRH and BarrCRH-like neurons (n = 3 per group) recorded in the same mouse showing their distinct spike waveforms and autocorrelations with a marked central valley feature indicating that they each represent a discriminated unit.

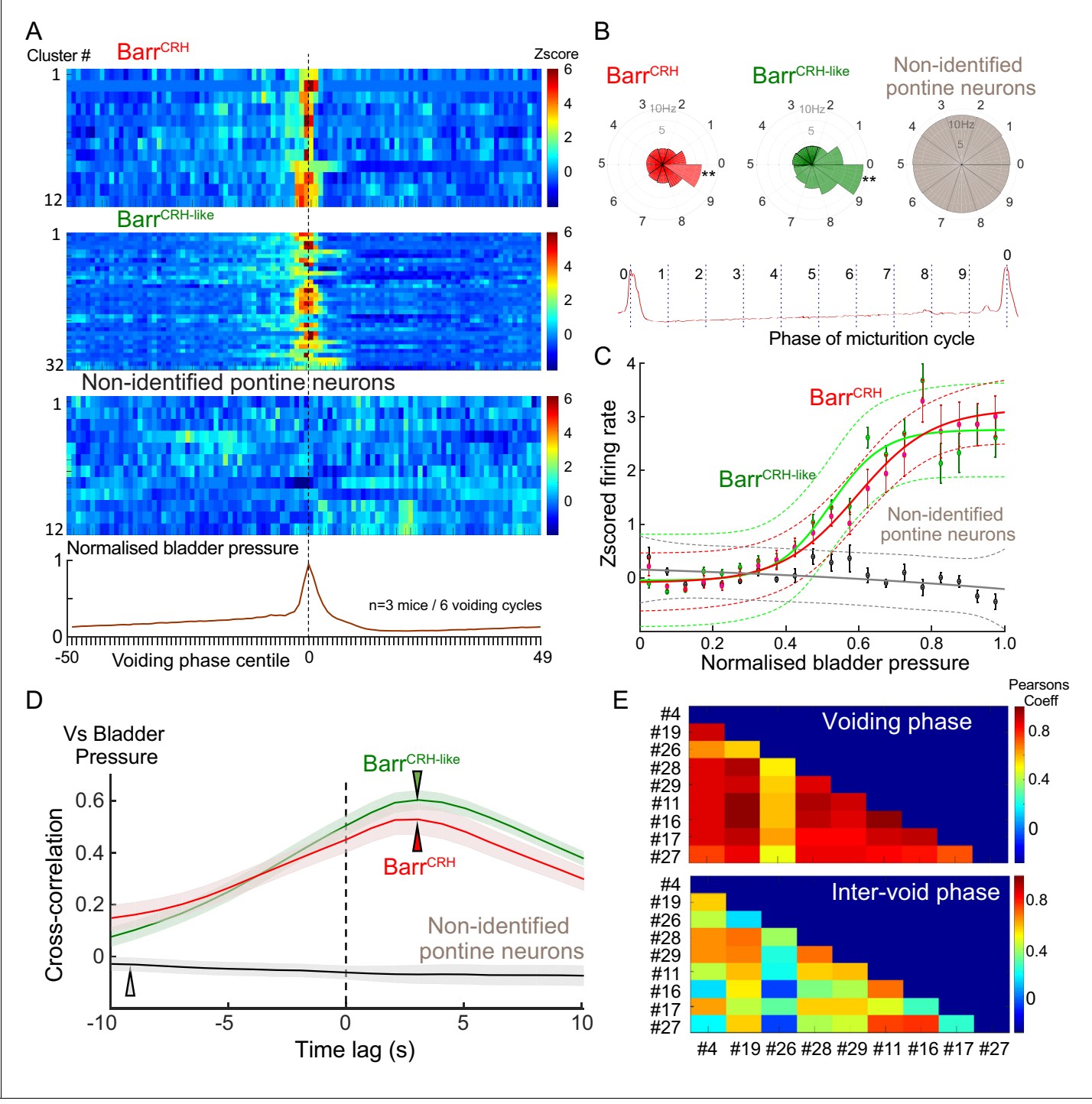

**Figure 10.** Population dynamics of Barr[CRH] and Barr[CRH-like] neurons. (**A**) Firing rate heat maps from probe recordings across mice (*n = 3*) with opto-identified Barr[CRH] neurons (*n = 12*, (respectively 8, 3 and 1 in each mouse)) and Barr[CRH-like] neurons (*n = 32*, (respectively 9, 19 and 4 in each mouse)) showed very similar patterns of firing in relation to the voiding cycle (shown below normalized for pressure and time across six cycles). (**B**) Rose plots of firing activity against phase of micturition cycle showing that both Barr[CRH] and Barr[CRH-like] neurons increase their firing in the phase decile leading up to the void unlike the unidentified neurons (**-P < 0.01, one-way ANOVA followed by Tukey-Kramer test). (**C**) Plotting the relationship between firing rate and normalized bladder pressure showed a graded sigmoid relationship with increased firing rate corresponding to higher bladder pressures. No such relationship was seen for the other neurons in the dorsal pons (dotted lines mark 95% CI of curves, bars SEM of firing rate) (**D**) The cross correlation between Barr[CRH] (and Barr[CRH-like]) neurons and bladder pressure was strongest at a lag of 3 s indicating that the bladder pressure follows the change in

*Figure 10 continued on next page*

*Figure 10 continued*

neuronal firing (shaded area marks SEM of mean cross correlation) (E) Colour plots of the Pearson's cross-correlation coefficient between pairs of the population of Barr$^{CRH}$ and Barr$^{CRH-like}$ neurons is consistently strongest in the voiding phase.

The online version of this article includes the following source data and figure supplement(s) for figure 10:

**Figure supplement 1.** Dynamics of Locus coeruleus neurons with respect to the micturition cycle.

**Figure supplement 1—source data 1.** Data for 'Dynamics ofLocus coeruleusneurons with respect to the micturition cycle'.

Barr$^{CRH}$ neurons that was evident in cross-correlograms (*Figure 9—figure supplement 1*). Both Barr$^{CRH}$ and Barr$^{CRH-like}$ neurons showed a clear temporal relationship to bladder pressure (*Figure 9C*) with their firing preceding and ramping up with the pressure during voiding.

For both the Barr$^{CRH}$ and Barr$^{CRH-like}$ neurons there was a strong sigmoid relationship between bladder pressure and neuronal firing (*Figure 10C*), which wasnot seen in the non-identified group of neurons. The directionality of this influence was investigated by examining the cross-correlation between firing rate and bladder pressure – this indicated that the increases in firing frequency (for both Barr$^{CRH}$ and Barr$^{CRH-like}$ neurons) preceded increases in bladder pressure by ~3 s for both sets of neurons (*Figure 10D*). These data indicated that the pattern of firing of both Barr$^{CRH}$ and Barr$^{CRH-like}$ neurons anticipated changes in bladder pressure as would be expected for a pre-motor population upstream of bladder parasympathetic neurons.

Our recording probe trajectory passes close to the locus coeruleus (LC) on its lateral edge raising the possibility that some of our recorded (non-optoidentified) neurons could be LC neurons, which have been shown in rats to increase their firing in anticipation of a void (*Manohar et al., 2017*) and as such could fall into the Barr$^{CRH-like}$ group. To examine this possibility, we made recordings from LC neurons under identical recording conditions in mice which had received injections of CAV2-PRS-ChR2-mCherry directly into LC, causing selective expression of ChR2 in LC cells only (*Li et al., 2016*). A total of 29 opto-identified LC neurons were recorded (n = 3 mice). They showed a characteristic pattern of spontaneous firing and a phasic burst of activity with a paw pinch. LC cells increased their firing around the void – a pattern that was evident in individual firing rate plots and in the z-scored firing heatmap (*Figure 10—figure supplement 1A and B*). However this peri-void activation was significantly less pronounced than the increase in firing seen in the Barr$^{CRH-like}$ (and Barr$^{CRH}$) neurons (*Figure 10—figure supplement 1C*).This suggests that the Barr$^{CRH-like}$ neurons are most likely to be part of the population of Barr$^{CRH}$ neurons, of which only a subset recorded by the probe are exposed to enough light to be formally opto-identified.

## Barr$^{CRH}$ neuronal excitability is not altered during the micturition cycle

Analysis of spontaneous Barr$^{CRH}$ firing rates over the micturition cycle shows a pattern of activity that is consistent with what would be expected for a high-fidelity controller of bladder pressure. However, this is at odds with our optogenetic activation findings. To resolve this discrepancy the relationship between cycle phase and the light-evoked Barr$^{CRH}$ activity and voiding was examined in more detail.

During all phases of the voiding cycle it was possible to opto-excite Barr$^{CRH}$ neurons (*Figure 11A*) and the increase in firing frequency in both absolute and relative terms was independent of the phase of the micturition cycle (*Figure 11B*, ranging from 22.4 ± 7.7 to 24.0 ± 6.5 Hz across micturition phases). These data indicate that the intrinsic excitability of the Barr$^{CRH}$ neurons does not vary across the micturition cycle and that augmentation of the bladder pressure responses (by 17.0 ± 3.9 fold) occurs downstream of the firing output from Barrington's nucleus.

To further explore this proposition, the relationship between spontaneous non-voiding contractions (sNVC) and Barr$^{CRH}$ neuronal firing was mapped. sNVC are defined as phasic increases of intravesical pressure seen during filling cystometry, not associated with passage of urine and have been seen in many studies of murine urodynamics (*Pavcovich and Valentino, 1995*; *Hou et al., 2016*; *Ito et al., 2017*; *Keller et al., 2018*; *Verstegen et al., 2019*). A burst of firing in the Barr$^{CRH}$ neurons preceded the sNVC by 1.5–3.0 s – suggesting that they were triggered by a signal from the pons (*Figure 11C and D*). However, there was only a weak relationship between the magnitude of each Barr$^{CRH}$ burst and the amplitude of the associated sNVC (see *Figure 11E*). A linear fit of these data indicates that an increase in burst size of 20 spikes (close to the maximum observed range) would

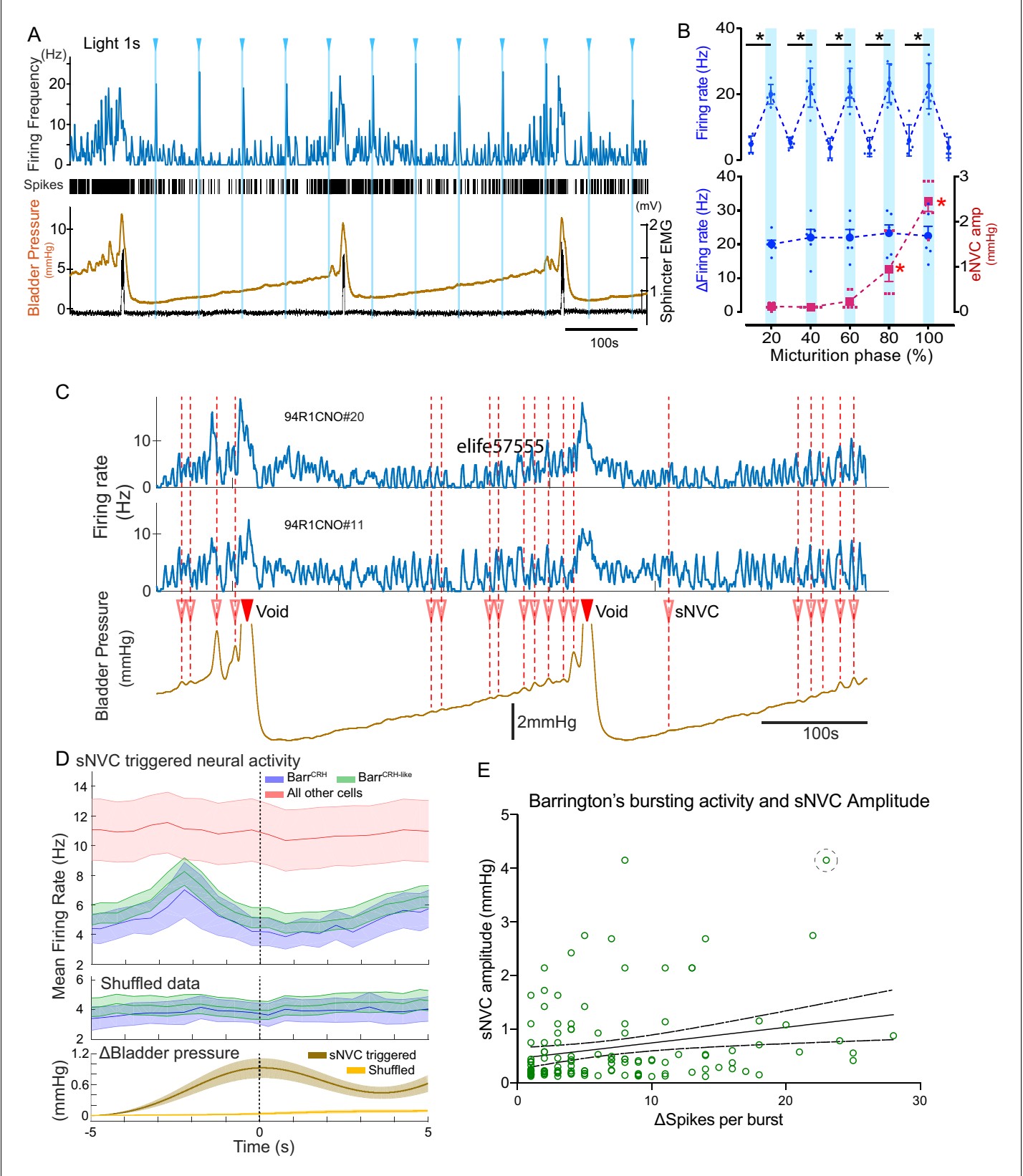

**Figure 11.** Barr[CRH] neuronal activity conditionally drives bladder pressure. (**A**) Optogenetic stimulation of Barr[CRH] neuron showing the transient increases in firing evoked by light pulses (1s × 465nm, pale blue lines) applied at different points of the micturition cycle. (**B**) Pooled data from Barr[CRH]

*Figure 11 continued on next page*

Figure 11 continued

neurons ($n$ = 6 across three mice) showing that there was no difference in firing (either the peak firing rate (upper) or the change in firing (lower, blue circles)) evoked by light across the phases of the micturition cycle. In contrast the amplitude of the eNVC (see *Figure 3*) increases markedly across the micturition cycle. (red squares) (**C**) Spontaneous NVCs were identified using a peak finding algorithm (amplitude 0.1–4 mmHg, green dotted circles) and were noted to be preceded by a burst of Barr^CRH activity. (**D**) Averaged firing rate plots of Barr^CRH and Barr^CRH-like neurons triggered off sNVCs (averaged bladder pressure trace at the bottom) showed a consistent burst of firing between 1.5–3 s before the onset of sNVCs (unlike the unidentified population). Note this relationship was not seen in the shuffled data. (Mean firing rates ± S.D, 0.5 s bins). (**E**) Linear regression showed the number of spikes in each Barr^CRH burst only showed a weak correlation (slope 0.03 mmHg/spike) with the amplitude of the following sNVC. This weak relationship was lost if a single outlier point was excluded (ringed). Source data in *Figure 11—source data 1*.

The online version of this article includes the following source data for figure 11:

**Source data 1.** Data for 'BarrCRH neuronal activity conditionally drives bladder pressure'.

only account for 0.5 mmHg difference in sNVC size (less than 20% of the observed range of amplitudes). Even this modest relationship was noted to be dependent upon a single outlier value of a large NVC occurring close to a void (circled). Again, this finding is consistent with Barr^CRH providing a trigger signal rather than a pre-motor drive which determines the amplitude of the bladder contraction.

## A spinal gate for the Barr^CRH drive is opened by bladder distention

This indicates a model of autonomous micturition where a spinal circuit gates the output to the bladder (shown schematically in *Figure 12A*). The Barr^CRH – parasympathetic - bladder afferent component of this circuit was modelled in NEURON using an existing preganglionic neuronal

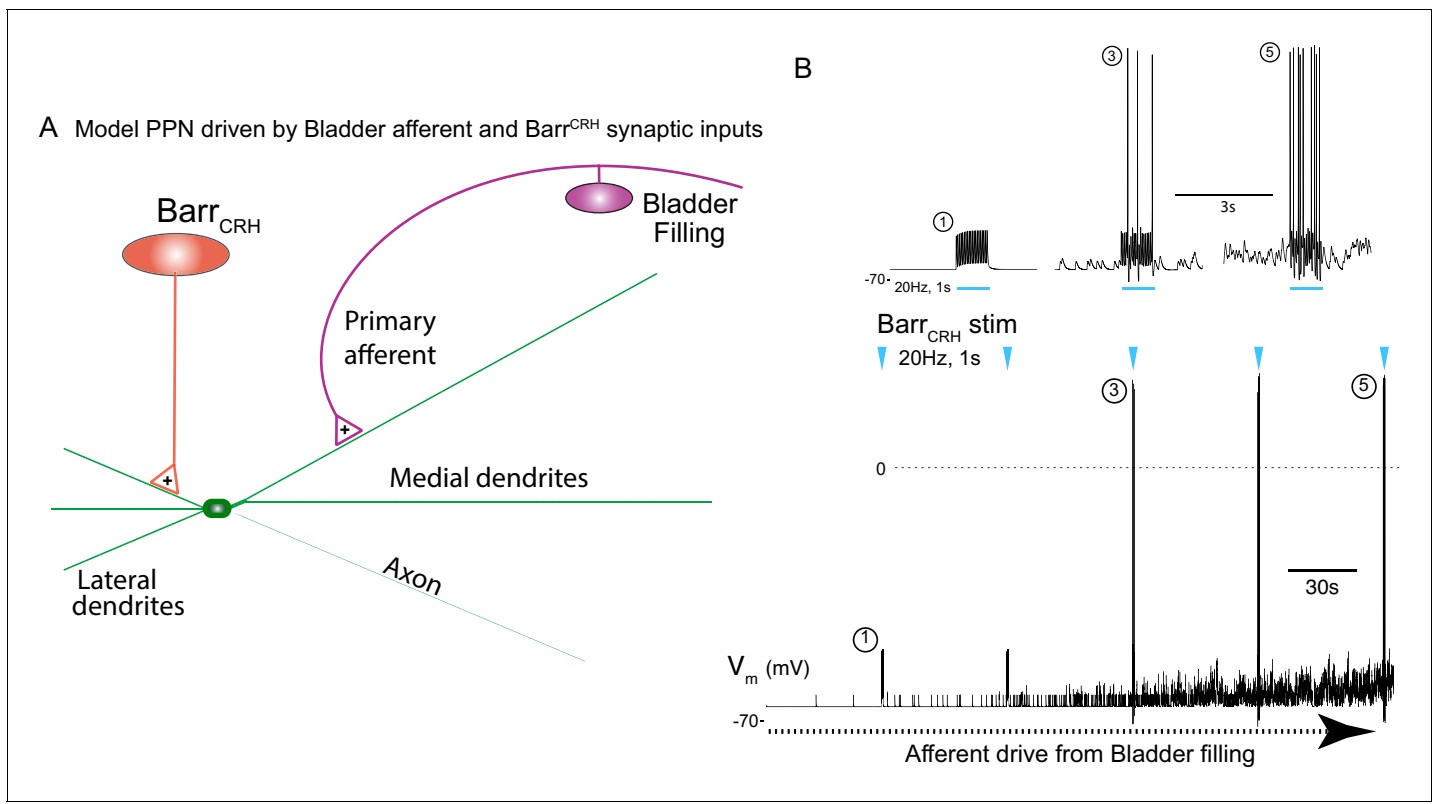

**Figure 12.** Integrative model of Barr^CRH drive to bladder parasympathetic neurons in micturition cycle. (**A**) Model of a parasympathetic preganglionic neuron (implemented in NEURON) with a synaptic drive from Barrington's nucleus and a second synaptic input from a bladder afferent neuron. (**B**) The model PPN is depolarised as the firing of the afferent neuron increases with bladder distension (afferent input from recordings *Ito et al., 2019*) producing an augmenting synaptic excitation (subthreshold for spike firing). Coincident 20 Hz stimulation of the Barr^CRH neuron (blue arrow heads, mimicking opto-activation) evokes no parasympathetic spike output at the start of the cycle but this increases to 8–10 Hz by the end of the cycle (traces shown on expanded timebase above).

model (*Briant et al., 2014*) and a combination of a fast, excitatory synaptic drive descending from Barrington's nucleus plus a bladder afferent synaptic drive (based on recordings of pelvic nerve afferents from *Ito et al., 2019*. The incrementing frequency of afferent drive, as the bladder fills, leads to summation and a maintained membrane depolarisation that increases parasympathetic excitability (*Figure 12B*).

The resulting output from the parasympathetic neuron when driven by Barr[CRH] (with a mimicked 20 Hz optogenetic drive) was strongly dependent upon the phase of the micturition cycle with a ~ 10 fold increase over the voiding cycle which closely parallels the experimental data. Note also that in the early phase of the voiding cycle the Barr[CRH] input is unable to evoke action potentials – thus producing 'failures'.

## An inferential model of autonomous micturition

The observations described above provided the basis for a new integrated model of the autonomous micturition cycle which incorporates the observed drive from Barr[CRH] neurons and the known afferent feedback from the bladder (*Figure 13A* and methods including a summary of the evidence supporting the model). This afferent feedback governs both the excitability of the spinal parasympathetic neurons (demonstrated in the NEURON model above) and the output of a synaptic generator driving Barr[CRH] activity. The resulting feedback loop (depicted schematically in *Figure 13B*) closely reproduces characteristic of the observed micturition cycle with graded NVCs, periodic voids and patterns of Barr[CRH] firing.

The varying excitability of spinal parasympathetic neurons is represented in the integrated model by a pressure-modulated logistic relationship which determines the change in bladder pressure generated from a given level of Barr[CRH] firing. The afferent drive also determines the probability of a high frequency Barr[CRH] discharge in a given epoch. The bladder pressure-dependent synaptic drive for Barr[CRH] is represented as a logistic relationship. This synaptic generator is commonly believed to be relayed via the PAG (*Drake et al., 2010*; *de Groat and Wickens, 2013*; *de Groat et al., 2015*) however, there is also evidence for direct spinal inputs to Barrington's nucleus (in the rat) that could also act as a generator (*Ding et al., 1997*; *Blok and Holstege, 2000*). In addition, elegant recent studies indicate there are also direct functional inputs from the cortex and hypothalamus (*Hou et al., 2016*; *Yao et al., 2018*; *Verstegen et al., 2019*).

This circuit organisation generates NVCs: dynamic perturbations whose magnitude and frequency increase with progress through the micturition cycle. As pressure increases these contractions become more frequent and higher in amplitude – eventually summating to cause sustained increases in bladder pressure. This in turn increases the rate of firing of Barr[CRH], making further contractions more likely, and shifting the system into a positive feedback loop in which pressure rapidly increases. A void occurs when the pressure reaches 15 mmHg which is presumed to be effected via a spinal mechanism and the micturition cycle restarts.

In line with experimental data, attenuation of the variance in Barr[CRH] firing (underpinning the NVCs) delays the time to void – indicating their importance in the process (*Figure 13—figure supplement 1A*). Similarly, augmenting the spinal parasympathetic sensitivity to the Barr[CRH] drive (as seen experimentally with intrathecal Astressin) increases the amplitude of the NVCs and shortens the inter-void interval (*Figure 13—figure supplement 1C*). We note that additional drive into the Barr[CRH] neurons (as is proposed to come from higher centres with voluntary voiding) would increase the variance and could trigger voiding earlier. This effect is demonstrated with the simulated optogenetic drive of Barr[CRH] neurons (20 Hz x 1 s, *Figure 13—figure supplement 1B*) which produces both failures, eNVCs and triggers voids earlier in the cycle than would otherwise have happened.

## Discussion

These findings indicate that Barr[CRH] neurons do play a critical role in micturition. However, the activity of Barr[CRH] neurons, is not a simple switch mechanism for voiding nor do they provide a direct drive to bladder pressure (as might be expected for an autonomic command neuron). Instead these neurons play a more nuanced, probabilistic role. Their influence on the bladder depends on the state of priming of the downstream parasympathetic motor circuit. This identifies the Barr[CRH] neurons as being the efferent limb of an inferential circuit that assays bladder state repeatedly during the storage phase of the cycle. When the threshold for voiding is reached, they generate a high-fidelity

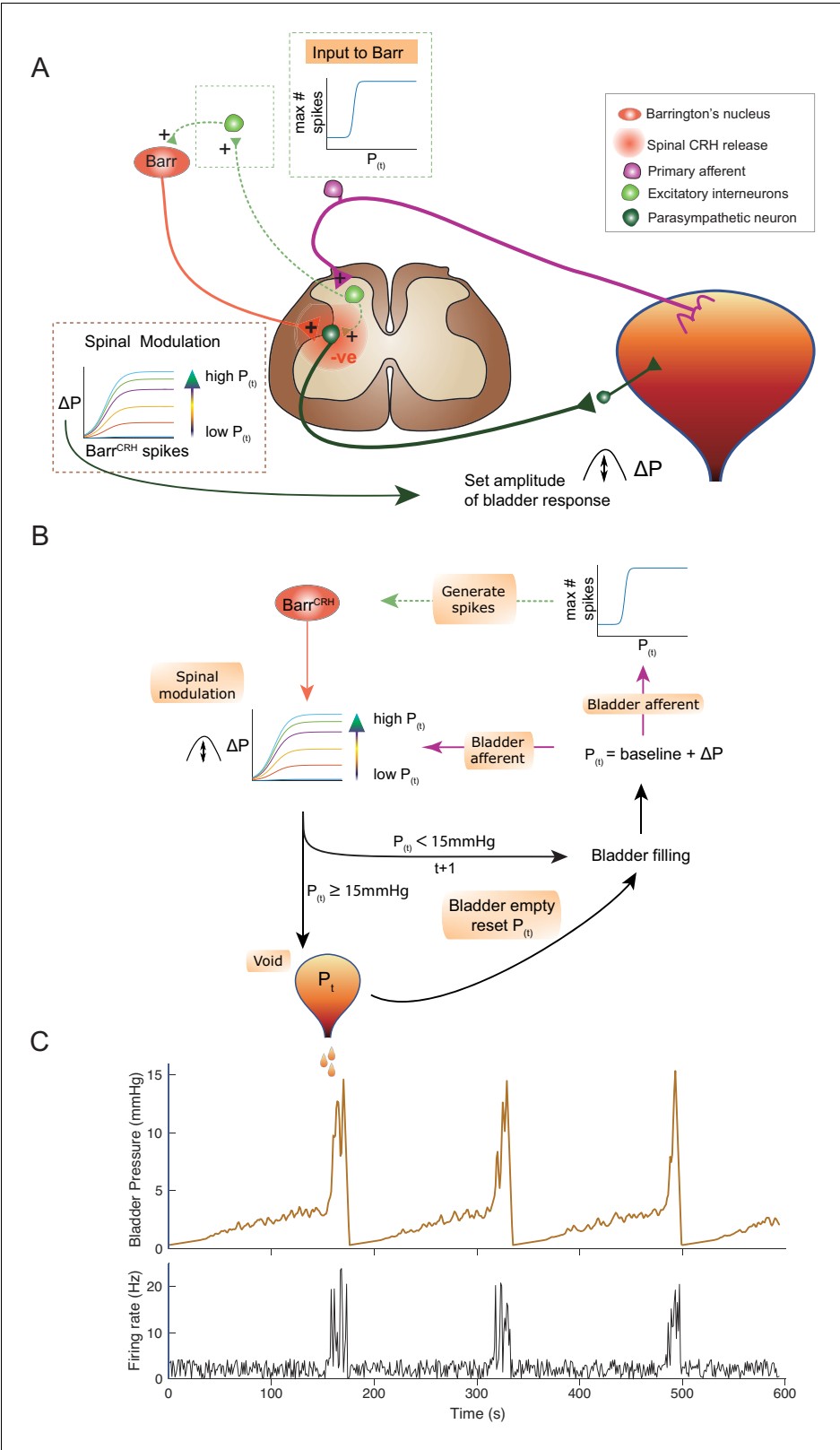

**Figure 13.** An inferential model of autonomous micturition. (**A**) Schematic of the descending input from Barrington's nucleus to the bladder parasympathetic neurons. The parasympathetic neurons receive excitatory input from bladder afferents – shown as being relayed via a segmental excitatory interneuron. Note that the Barr$^{CRH}$ neuron has both a fast, excitatory transmitter (presumed glutamate) as well as an inhibitory action mediated by spinally released CRH – possibly acting via local inhibitory interneurons (not shown). The inset boxes show the logistic relationships linking

*Figure 13 continued*

activity of Barr[CRH] neurons and spinal excitability to the current bladder pressure (from model in B). (**B**) Flow chart showing processing steps in inference model of micturition. (**C**) Output from the model showing incrementing bladder pressure with NVCs over three micturition cycles with the associated Barr[CRH] firing that generates the NVCs and the voids.

The online version of this article includes the following figure supplement(s) for figure 13:

**Figure supplement 1.** Inferential micturition model recapitulates observed behaviour.

motor signal through a positive feedback loop that drives the bladder contraction required for voiding.

This operating principle fits with a modular hierarchical hypothesis for the organisation of the micturition circuit (*Fowler et al., 2008*; *Drake et al., 2010*; *de Groat and Wickens, 2013*) with a primary spinal circuit providing a basic functionality, evident in the neonatal rodent (*Kruse and De Groat, 1990*; *de Groat, 2002*; *Zvarova and Zvara, 2012*) and indeed other mammals including humans, that has little context-sensitive control. The timing of micturition in immature rodents is often triggered by maternal stimulation of the perineum (although interestingly this is unsuccessful if applied when the bladder is <50% full *Zvarova and Zvara, 2012*). With development, the spinal micturition mechanism is believed to fall progressively under the descending control of Barrington's nucleus (both for voluntary and autonomous voiding). We suggest that such descending control provides an internalised signal, replacing the need for additional external peripheral sensory input, to trigger the void. Dysfunction of this descending control system, as is seen following spinal cord injury, results in a loss of voluntary control and initially in a complete loss of continence, but this tends to be restored as the spinal micturition reflex re-emerges (albeit in a poorly co-ordinated manner). This situation was mimicked experimentally herein by the chemogenetic inhibition of Barr[CRH] neurons – leading to a prolongation of the inter-void interval and retained volumes with progressive bladder distension – indicating that this is a necessary and critical component of the micturition circuit.

We used cystometry in anaesthetised mice to examine the role of Barr[CRH] neurons specifically in the core processes of autonomous micturition in the absence of behavioural influence. This has produced a number of different findings from previous optogenetic studies of the role of Barr[CRH] neurons in micturition (*Hou et al., 2016*; *Keller et al., 2018*; *Verstegen et al., 2019*) which we have ascribed to the contrast between volitional and autonomous micturition behaviour. However, an important caveat is that anaesthetic agents by definition alter neuronal function, typically suppressing activity, and they have been found to affect aspects of murine micturition by a number of authors (see review *Ito et al., 2017*). Urethane has been adopted by many research groups as the agent producing the least autonomic suppression and for its stable plane of anaesthesia. The cystometric profile and the characteristic activity of the EUS seen in this study is akin to that seen in awake mice and in decerebrate arterially perfused preparations in the absence of anaesthesia (*Ito et al., 2017*; *Ito et al., 2018*). Additionally, the fact that we find an apparent increase in the role of Barr[CRH] neurons argues against urethane anaesthesia accounting for the differences between our findings and those in conscious mice. This remains to be definitively tested and we hypothesise that the role of Barr[CRH] neurons in autonomous voiding may be best demonstrated in sleeping rather than conscious, behaving mice.

Rodents also use social urine scent marking, for example male mice use strategic urine 'spotting' to express their dominance and territorial ownership (*Desjardins et al., 1973*; *Maruniak et al., 1974*). Although autonomous micturition and scent marking with urine are related processes, likely with some shared physiology, there are also differences in the patterns of urination with greater frequency (>10 urine spots per minute *Keller et al., 2018*) and accordingly smaller volumes in each urine spot compared to a primary void (typically 80–120 ul) (*Yu et al., 2014*; *Bjorling et al., 2015*; *Hill et al., 2018*; *Ito et al., 2018*). A role has recently been described for Barr[ESR1] neurons in social urine spotting evoked by female urine (*Keller et al., 2018*). These Barr[ESR1] neurons preferentially target a spinal inter-neuronal circuit that is proposed to be involved in generating the bursting drive to the EUS as well as causing a bladder contraction. In contrast, in the same study, the Barr[CRH] neurons were reported to be relatively ineffective in generating voids under similar conditions (without active filling of the bladder) (*Keller et al., 2018*).

A similar conclusion was reached by Verstegen and colleagues who compared the effect of activating Barr[CRH] neurons with the activation of all the glutamatergic neurons in Barrington's nucleus (including the ESR1 neurons) and found that global glutamatergic activation produced obligatory voiding with the characteristic of incontinence. In contrast activation of Barr[CRH] neurons only sporadically produced co-ordinated voiding with a delay and only with 6% of activations although this was a little higher at 17% in anaesthetised mice (*Verstegen et al., 2019*). This evidence led both groups to conclude that Barr[CRH] neurons only played a minor supporting, augmentative role in the generation of voids (*Pavcovich and Valentino, 1995*; *Hou et al., 2016*; *Ito et al., 2017*; *Keller et al., 2018*; *Verstegen et al., 2019*). This is somewhat surprising given that the majority of spinally projecting Barr neurons are CRH positive (*Verstegen et al., 2017*) raising the question of why they have such an apparently minor role in voiding. With hindsight, a clue to this puzzle may have been offered by the finding of *Verstegen et al. (2019)* that genetic, diphtheria toxin-mediated, ablation of the Barr[CRH] neurons produced a phenotype of increased voided volume in awake mice and markedly delayed voiding during cystometry in anaesthetised mice.

Our study shows that Barr[CRH] neurons can effectively trigger co-ordinated voiding with the characteristic pattern of external urethral sphincter bursting when the bladder is sufficiently filled. This triggering is not just a simple consequence of the pressure rise produced by Barr[CRH] neuronal activation (ie a switch) - as prolonged activation of Barr[CRH] neurons leads to increased voiding frequency with a lowered threshold for voiding (and chemogenetic inhibition has the opposite effect). Indeed, the concept of the existence of a simple pressure threshold for triggering a void is challenged by the observation that large spontaneous non-voiding contractions occurred just prior to a void that exceed any estimated threshold for a subsequent void. Rather we believe that the drive from Barr[CRH] needs to be integrated at a spinal level with feedback from bladder afferents in order to generate a complete void – and thus the probability of evoking a void depends on the degree of bladder filling. When the bladder is partially filled, activation of the Barr[CRH] neurons evokes bladder contractions (eNVC) without any sphincter activity, suggesting that the drive to the sphincter is also dependent on the state of a downstream pattern generator and is not directly engaged by the firing of Barr[CRH] neurons at all stages of the micturition cycle. Our preliminary experiments found no evidence for the involvement of Barr[CRH] neurons in the control of the distal colon, suggesting they are bladder specific.

Given the mechanistic differences between the processes of autonomous micturition and voluntary scent marking in males, it is quite likely that there are distinct circuit drives for each type of urination. This would be consistent with the proposition that there are parallel pathways from Barrington's nucleus to the downstream spinal pattern generators: the Barr[ESR1] neurons driving spotting behaviour which can be triggered irrespective of the degree of fullness of the male mouse bladder; and the other mediated by Barr[CRH] neurons which conditionally requires the bladder to be distended before a void can be generated. This may also explain why chemogenetic inhibition of the Barr[ESR1] but not Barr[CRH] neurons blocked spotting whereas similar inhibition of Barr[CRH] neurons inhibited autonomous micturition in the current study. Our study did not set out to identify gender differences in Barr[CRH] neuronal function in micturition and we included mice of both sexes. Although female mice had smaller bladder capacities along with larger basal and micturition pressures on CMG there were no differences in their other parameters and their EUS-EMG trace was qualitatively similar in pattern during voiding. Equally both sexes showed similar responses to activation/inhibition of Barr[CRH] neurons which presumably relates to underlying commonalities in the processes of autonomous micturition.

In this context it is also important to acknowledge that the pattern of micturition varies to a degree across species and humans and cats do not show the same 'squirting' behaviour as their EUS relaxes to allow voiding – this likely involves some distinctive neuronal circuitry (*Fowler et al., 2008*). However, there are also many similarities including, surprisingly, in the time taken to void irrespective of body size (*Yang et al., 2014*) indicating that many of the fundamental principles of operation are conserved. Importantly, lesions or inhibition of Barrington's nucleus abolishes voiding in multiple species including humans, cats, rats and mice consistent with it being a core circuit component reviewed in *Verstegen et al. (2017)*.

The pattern of firing activity seen here in optogenetically-identified Barr[CRH] neurons in anaesthetised mice is similar to that previously noted in recordings from Barrington's nucleus in the conscious rat (*Manohar et al., 2017*) and is also reminiscent of a subset of the neurons identified in

anaesthetised or decerebrate rats and cats that showed a ramping activity with voiding (*de Groat et al., 1998*; *Sugaya et al., 2003*; *Tanaka et al., 2003*; *Sasaki, 2005b*; *Sasaki, 2005a*). The bursting activity seen in the Barr$^{CRH}$ recordings clearly precedes the changes in bladder pressure (and with a similar lag to bladder pressure response to that found from optogenetic activation of Barr$^{CRH}$ neurons) indicating that they are driving rather than responding to the changes in pressure. The Barr$^{CRH}$ neurons showed a pattern of spiking activity that is also consistent with that noted from the Ca$^{2+}$ imaging recordings seen with fibre photometry in mice (*Hou et al., 2016*; *Keller et al., 2018*; *Verstegen et al., 2019*) although we can see both the temporal precedence and that this activity decays promptly at the end of the void. It is also worth noting that in none of these recordings (from Barr$^{CRH}$ or indeed any of the neurons in the vicinity) was any pattern of activity seen that resembled the high frequency bursting of the urethral sphincter seen in mice and rats – to date such activity has never been observed in any of the recordings from Barrington's nucleus which is consistent with the idea that it is generated from a spinal motor pattern generator such as the LSCC (*Chang et al., 2007*).

The probabilistic nature of the influence of Barr$^{CRH}$ neurons on bladder pressure seems initially at odds with the clear relationship between their activity and bladder pressure noted herein (and previously in the E2 class of neurons recorded in rat Barrington's nucleus [*Tanaka et al., 2003*] and also in fibre photometry recordings *Hou et al. (2016)*; *Verstegen et al., 2019*). However, this relationship only holds in the late stage of the micturition cycle when Barr$^{CRH}$ neurons do indeed act as a tightly-coupled, direct command neuron. This is not the case during the early phases of the micturition cycle when there is a weak relationship between the activity of Barr$^{CRH}$ neurons and the bladder pressure, in the extreme case leading to 'failures' of stimulation to evoke any contraction. Our recordings indicate that this happens downstream of Barr$^{CRH}$ at a spinal level, as the ability to optogenetically drive Barr$^{CRH}$ is unchanged and spinal activation of Barr$^{CRH}$ axons also shows failures. We propose a model for this action that integrates an incrementing and summating, but still sub-threshold, afferent drive from the bladder to PPN that enables a phasic burst of activity from Barr$^{CRH}$ to generate progressively larger numbers of action potentials and hence contraction when the bladder is sufficiently filled. In support of this idea, previous electrical stimulation studies of Barrington's nucleus in the rat indicated that the degree of excitation of the parasympathetic motor outflow to the bladder was strongly dependent upon the degree of bladder filling (*Noto et al., 1991*). The mechanisms enabling such priming of the parasympathetic control circuit will merit further investigation at a spinal level.

The inhibitory action of CRH released from Barr$^{CRH}$ neurons on micturition at a spinal level initially appears counterintuitive given the overall excitatory effect of Barr$^{CRH}$ neurons (mediated via fast glutamatergic signalling, *Hou et al., 2016*) on micturition and the known excitatory effects at a cellular level of CRH receptor activation (*Lovejoy et al., 2014*). However, a similar inhibitory spinal action of CRH on micturition has been reported (*Pavcovich and Valentino, 1995*; *Kiddoo et al., 2006*; *Wood et al., 2013*). This inhibition may act to suppress the segmental excitatory activity in the spinal parasympathetic circuit. This process may also be involved in the transition from the immature spinal voiding circuit in the neonate that is supplanted by the top down requirement for Barrington's nucleus signals. The mismatch between the differential time-course of action of CRH-mediated inhibition (metabotropic) and the fast, glutamatergic excitation (ionotropic) may enable the initial rapid excitation of parasympathetic preganglionic neurones at the onset of voiding, but may also in turn act to help terminate voids and facilitate the unopposed relaxation of the bladder. The spinal mechanism of CRH actions at a spinal circuit level constitutes an intriguing target for therapeutic intervention potentially allowing modification of the gain of the micturition reflex in disease states.

We noted that bursts of activity in Barr$^{CRH}$ neurons precede both voids and also NVCs. There has been considerable debate about the origin of NVCs with respect to whether they are intrinsically generated by the bladder and their functional significance although there is a suggestion that they provide a means to infer the degree of bladder filling (*Drake, 2007*). An increase in their frequency and amplitude has been linked to diseases of the LUT (*Vahabi and Drake, 2015*) and also with loss of descending control from the brainstem (*Sadananda et al., 2011*) which could conceivably be related to the loss of CRH-mediated spinal inhibition. There is also evidence of their peripheral generation by the bladder early in development which becomes less coordinated in adult bladder (*Kanai et al., 2007*). We provide evidence that NVCs are generated by the 'noisy' probabilistic drive from Barrington's nucleus that repeatedly assesses the status of the spinal circuit during each

micturition cycle and that the magnitude of the bladder pressure response reflects the phase of the micturition cycle. The resulting afferent signal provides an active way of inferring the degree of bladder fullness (analogous to the 'sampling' that assesses rectal fullness *Rao, 2004*) and could prime the neural control circuits and indeed could conceivably provide a stream of information that may enable a conscious awareness of bladder fullness and the ability to make volitional predictions about the need to void. We also note the homology with the development of other motor systems where spontaneous motor activity, initially generated in the periphery, becomes progressively embedded centrally as motor representations in the nervous system with developmental maturation (*Llinás, 2001*).

Our postulated model of such a circuit organisation with afferent feedback from the bladder both priming the spinal parasympathetic motor circuit and also determining the magnitude of the drive from Barr[CRH] neurons (perhaps via integration by the PAG which functions as a probabilistic firing switch) recapitulates many of the observed features of autonomous micturition. The generation of NVCs provides inference about the degree of bladder fullness and the afferent signal advances the progression through the cycle. The spinal priming mechanism enables a regenerative burst of activity from Barr[CRH] to drive the voiding contraction and the modelled release of spinal CRH that follows such a large discharge serves to reset the spinal circuit enabling passive filling to resume. A feature of this circuit organisation is that a direct volitional drive to Barr[CRH] from cortex as recently reported (*Yao et al., 2018*) would not be subject to the probabilistic firing switch at a supraspinal level and could therefore trigger a void earlier in the cycle if behaviourally appropriate albeit still contingent on the priming status of the spinal gate. Hypothetically, a parallel synaptic drive from the Barr[ESR1] neurons (*Keller et al., 2018*) that was stronger than the Barr[CRH] neurons could also generate parasympathetic activity without requiring co-incident afferent activity – hence bypassing the gate to produce urine 'spotting' on behavioural demand.

On this basis we conclude that the Barr[CRH] neurons form a key component of the micturition circuit that generate a pre-motor drive to the bladder late in the cycle. The recording and stimulation data suggest that this drive is not generated by a burst generator residing within this cell population but is a product of the integration of inputs from both bladder sensory afferents and upstream centres such as the PAG but also including hypothalamus and motor cortex (*Hou et al., 2016*; *Yao et al., 2018*). We predict that failures of control at this key integrating locus are likely to be involved in both acute disorders of lower urinary tract function such as retention as well as in chronic diseases like nocturnal enuresis, detrusor-sphincter dyssynergia and overactive bladder syndrome where there is dysregulation of detrusor contractions and sphincter relaxation.

# Materials and methods

### Key resources table

| Reagent type (species) or resource | Designation | Source or reference | Identifiers | Additional information |
|---|---|---|---|---|
| Strain, strain background (*Mus musculus*) | CRH[Cre] | Jax Laboratory (*Taniguchi et al., 2011* DOI: 10.1016/j.neuron. 2011.07.026) | JAX#012704 RRID:IMSR_JAX:012704 | Male and Female Cre recombinase targeted to CRH locus. |
| Genetic reagent (Adeno-associated viral vector) | AAV-EF1α-DIO-hChR2(H134R)-mCherry | University of North Carolina vector core (*Gradinaru et al., 2007* DOI: 10.1523/JNEUROSCI. 3578–07.2007) | | Optogenetic transduction |
| Genetic reagent (Adeno-associated viral vector) | AAV-hSyn-DIO-hM4Di-mCherry | Addgene (*Krashes et al., 2011* DOI: 10.1172/JCI46229) | Plasmid #44362 | Chemogenetic transduction |
| Genetic reagent (Canine Adenoviral vector) | CAV-PRS-ChR2-mCherry | Plateforme de Vectorologie de Montpellier (*Li et al., 2016* DOI: 10.1016/j.brainres. 2016.02.023) | CAV-PRS-ChR2-mCherry | www.pvm.cnrs.fr/ plateau-igmm/ |

*Continued on next page*

*Continued*

| Reagent type (species) or resource | Designation | Source or reference | Identifiers | Additional information |
|---|---|---|---|---|
| Antibody | Anti-mCherry (Rabbit polyclonal) | Biovision | 5993, RRID:AB_1975001 | 1:4000 |
| Antibody | anti-ChAT (Goat polyclonal) | Millipore | Cat# AB144P RRID:AB_90650 | 1:250 |
| Antibody | anti-TH (Sheep polyclonal) | Millipore | Cat# AB1542 RRID:AB_90755 | 1:1000 |
| Chemical compound, drug | Urethane (ethyl carbamate) | Sigma | #U2500 | Anaesthetic |
| Chemical compound, drug | Clozapine-N-Oxide | Tocris | 4936 | DREADD ligand |
| Chemical compound, drug | Astressin | Sigma | A4933 | CRH antagonist |
| Software, algorithm | Parasympathetic preganglionic neuron model | NEURON, (based on *Briant et al., 2014*, DOI: 10.1152/jn.00350.2014) | | Available from data.bris.ac.uk (DOI: 10.5523/bris.20l920gl27 ufi204brn8ilonsf) |
| Software, algorithm | Autonomous micturition model | MATLAB code | | Available from data.bris.ac.uk (DOI: 10.5523/bris.20l920gl27 ufi204brn8ilonsf) |
| Software, algorithm | Analysis scripts for single cell recordings. | MATLAB code | | Available from data.bris.ac.uk (DOI: 10.5523/bris.20l920gl 27ufi204brn8ilonsf) |
| Software, algorithm | Clustering of multisite probe recordings. | Kilosort (*Pachitariu et al., 2016*) | | papers.nips.cc/paper/ 6326-fast-and-accurate- spike-sorting-of-high-channel -count-probes-with-kilosort |
| Software, algorithm | Cluster curation. | Phy (*Rossant et al., 2019*) | | github.com/kwikteam/ phy-contrib |
| Software, algorithm | Bladder pressure and EUS data acquisition/analysis. | Spike2 (Cambridge Electronic Design) | | |

## Experimental model and subject details

### Mice

All experiments and procedures conformed to the UK Animals (Scientific Procedures) Act 1986 and were approved by the University of Bristol Animal Welfare and Ethical review body and performed under licence (PPL3003362). Mice were group housed, with food and water available ad libitum and on a 12 hr/12 hr light/dark cycle.

Gene expression was restricted to Barr[CRH] neurons using knock-in mice (of both sexes aged 3–8 months old) with an internal ribosome entry site (ires)-linked Cre-recombinase gene downstream of the CRH locus (CRH[Cre] mice (*Taniguchi et al., 2011*; *Chen et al., 2015*), Jax Laboratory #012704).

### Quantification and statistical analysis

All data are presented as mean ± SEM (unless otherwise specified). Sample size was estimated from experience and is similar to other published studies (*Ito et al., 2019*; *Hou et al., 2016*; *Verstegen et al., 2019*).

Statistical tests used are specified in Figure legends and in the main text (see Results). Differences were considered significant at p<0.05. All experiments contain replications of the same experimental paradigm across different litters of animals and experimental runs. The number of replications (n) equals the number of mice for bladder pressure recordings, and/or the number of cells for electrophysiological experiments (as stated the relevant Figure legends/main text). Mice of either sex were allocated to experiments from the breeding colony as available. Blinding of the experimenter to

drug was used for the chemogenetic experiments, however no blinding was possible for the optogenetics/cell recording studies.

## Viral vectors

The serotype two recombinant AAV-EF1α-DIO-hChR2(H134R)-mCherry (*Nagel et al., 2005*; *Gradinaru et al., 2007*) ($1.6 \times 10^{12}$ viral genomes/ml) used for optogenetic activation experiments was obtained from University of North Carolina vector core facilities (a gift from Karl Deisseroth). The serotype 2 AAV-hSyn-DIO-hM4Di-mCherry (*Krashes et al., 2011*) ($7 \times 10^{12}$ vg/ml) used for chemogenetic inhibition experiments was obtained from Addgene (a gift from Brian Roth). To target the locus coeruleus selectively for optogenetic activation we used CAV-PRS-ChR2-mCherry ($2 \times 10^{11}$ pp/ml; *Li et al., 2016*).

## Stereotaxic intracranial injections to Barrington's nucleus

To target Barr^CRH neurons, homozygous CRH^cre mice were anaesthetised with ketamine (70 µg/g) and medetomidine (0.5 µg/g) and placed in a small animal stereotaxic frame (Kopf, USA) with a drill-injection robot attachment (Neurostar, Germany). After exposing the skull under aseptic conditions, a small burr hole was drilled and AAVs were injected (200 nl x three injections per side) unilaterally or bilaterally through a pulled glass pipette at a rate of 100 nl/min. Injection coordinates for Barrington's nucleus were 5.3 mm posterior to bregma, 0.70 mm lateral and 3.25, 3.5 and 3.75 mm below brain surface. Injections for Locus coeruleus were identical but targeted 0.8 mm lateral. After surgical procedures, all mice were returned to their home cage for at least 21 days for recovery to maximise protein expression.

## Optogenetic activation

To target Barrington's nucleus, adult CRH^Cre mice had injections of AAV-DIO-ChR2-mCherry ($1.6 \times 10^{12}$ vg/ml) or AAV-DIO-hM4Di-mCherry (as control). To target Locus coeruleus, adult C57Bl/6 mice had injections of CAV-PRS-ChR2-mCherry ($2 \times 10^{11}$ pp/ml). Mice were used in experiments at least 3 weeks after vector injections. They were anaesthetised with urethane and prepared for cystometry as described below. Light from a 465 nm LED (Plexon, Dallas USA) was delivered in pulses with a maximum duty cycle of 50%. The light train was delivered once every 60 s for fixed-interval stimulation, or at randomised intervals between 30 s and 90 s. The light power exiting the fibre tip was set at approximately 10 mW and was measured before and after each experiment. For unilateral opto-activation light was delivered via a tapered optical fibre (Lambda-B, 0.39NA, 17 mm long, 1.2 mm emitting length, Optogenix, Italy) with the fibre lowered down the original vector injection track. For bilateral simultaneous opto-activation a dual fibre implant was used (DFC 200/250–0.66 15 mm DF1.4 C60, Doric, Canada) and coupled via a dual fibre optic cable to two separate LEDs.

For light delivery to the spinal cord, soft tissue was removed between T11 and T12 vertebral spines after skin incision. The exposed spinal cord was illuminated using a 473 nM laser (PhoxX, Omicron, Germany) via a bare ended fibre (Thorlabs, 400 µm) positioned above the cord and delivered in 20 ms pulses at 20 Hz for 5 s or a prolonged pulse of 1000 ms. The light power at the fibre tip was $29 \pm 0.3$ mW.

## Chemogenetic inhibition

To inhibit Barr^CRH neurons, CRH^cre mice were bilaterally injected with AAV-DIO-hM4Di-mCherry ($7.0 \times 10^{10}$ vg/ml) into Barrington's nucleus (as described above) and allowed at least 3 weeks of recovery (control mice had AAV-DIO-ChR2-mCherry injected). They were anaesthetised with urethane and prepared for cystometry as described below. Intraperitoneal CNO (5 mg/kg, 1 mg/ml stock) or saline (as control) was applied after obtaining >five baseline micturition cycles. In an initial set of experiments, saline was continuously infused to the bladder around the time of CNO injection to investigate the effects on mice micturition. Subsequently, to determine the CNO effect on threshold for micturition, a cyclical infuse and hold protocol was adopted whereby saline infusion was stopped at the threshold for voiding and then held at that volume for 10 min or until a void occurred before emptying the bladder and restarting the infusion phase.

## Cystometry, Electromyography and distal colonic manometry

Mice were anaesthetised with urethane (0.8–1.2 mg/kg) and the bladder was exposed via a 2 cm midline abdominal incision. A flanged catheter (PE50) was secured with a purse-string suture into the bladder and connected to a syringe pump and pressure transducer. The infusion rate was adjusted on an individual mouse basis (10–40 µl/min) to produce an equivalent proportionate speed of fill to threshold for voiding (typically 600 s) taking account of differing bladder volumes of the mice. External urethral sphincter (EUS) was recorded with insulated stainless steel wires, bared at the tip (0.075 mm, AISI316 Advent) inserted through a 30 G needle bilaterally into the EUS just proximal to the pubic symphysis. A balloon catheter (2.5 mm diameter x 12 mm when fully distended, Medtronic Sprinter) was inserted into distal colon and the tip of balloon was placed 40 mm from the anus. To monitor colonic pressure the balloon catheter was filled with distilled water.

Once a regular rhythm of micturition cycles was established (typically ~1 hr after starting saline infusion into the bladder) then the following variables were measured (and averaged over at least three voiding cycles (see *Figure 1—figure supplement 2*):

- Basal pressure was taken as the lowest bladder pressure reached after a void.
- Voiding threshold was the bladder pressure when the EUS-EMG started bursting, indicating the initiation of voiding.
- Micturition pressure was the peak bladder pressure achieved during voiding (bursting phase of the EUS-EMG).
- Non-voiding contractions (NVCs) were identified as discrete increases in bladder pressure (>0.1 mmHg) observed during the filling phase in voiding preparations.
- Bladder compliance was defined as bladder capacity / (threshold - basal pressure) (µl/mmHg) during filling.

## Pithed, Decerebrate Arterially-Perfused mouse (DAPM) preparation

The pithed DAPM preparation was used to examine the influence of bladder filling on pelvic nerve stimulation-evoked bladder contractions. The methods were as previously described (*Ito et al., 2018*; *Ito et al., 2019*) but in brief, mice were terminally anaesthetised with isoflurane, disembowelled through a laparotomy and the bladder was cannulated. The mouse was then cooled, exsanguinated, decerebrated and its spinal column was pithed to remove all central neuronal control. It was then moved to a recording chamber, perfused through the heart with warm (32 ℃) Ringer's solution (composition (mM): NaCl (125), NaHCO$_3$ (24), KCl (3.0), CaCl$_2$ (2.5), MgSO$_4$ (1.25), KH$_2$PO$_4$ (1.25); glucose (10); pH 7.35–7.4 with 95% O2/5% CO2). Ficoll-70 (1.25%) was added as an oncotic agent to the perfusate. The flow rate was adjusted (from 15 to 20 ml/min) to achieve a perfusion pressure of 50–60 mmHg. The pelvic nerve was identified, traced proximally and cut allowing the distal end to be aspirated into a bipolar stimulating electrode. Stimuli (10V, 1 ms, 4–10 Hz for 3 s) were applied to the nerve. The bladder was filled with saline to perform cystometry as above with filling limited to a ceiling pressure of 15 mmHg. The effect of pelvic stimulation on bladder pressure was examined with different degrees of bladder filling (0–70 µl).

### Extracellular recordings and signal acquisition

Recordings were made from Barrington's nucleus and the locus coeruleus in urethane anaesthetised mice using a 15 µm thick silicone probe with 32 channels (NeuroNexus, Model: A1 × 32-Poly3-10mm-25 s-177-A32). For recordings in Barrington's nucleus, the recording probe was lowered down a track using the same co-ordinates as the vector injection. An optical fibre was lowered on an intersecting track to target the nucleus from a caudal vector (bregma –8.8, ML 0.7 or 0.8 and 4.9 mm deep on an angle of 45° to the vertical). For recordings in LC, an identical configuration was used, but with ML coordinate 0.8 mm. Each channel (177 µm$^2$) was spaced from the neighbouring channels by 50 µm. A reference electrode (Ag/AgCl) was inserted into the scalp. The probes were connected to an amplifier-digitising headstage (INTAN, RHD2132). The signals were amplified and filtered (100 Hz-3 kHz) and digitized at 30 kHz before being processed and visualised online within the Open Ephys system.

## Anatomical tracing studies

To investigate the Barr[CRH] projection to the spinal cord, Cre-dependent AAV (AAV- EF1α-DIO-ChR2-mcherry) was unilaterally injected to Barrington's nucleus in CRH[CRE] mice. After a minimum of four weeks the mice were killed, and perfusion fixed for immunohistochemistry. To examine the Barr[CRH] projection into spinal cord, 40 μm transverse sections were taken from T11 to S2, processed for mCherry and Choline acetyltransferase immuno- (to demarcate motoneurons) followed by confocal imaging (detailed below).

## Immunohistochemistry of brain and spinal cord

Mice were killed with an overdose of pentobarbital (20 mg per mouse, i.p; Euthetal, Merial Animal Health) and perfused trans-cardially with 4% formaldehyde (Sigma) in phosphate buffer (PB; pH 7.4, 1 ml/g). The brain and spinal cord were removed and post-fixed overnight before cryoprotection in 30% sucrose in phosphate buffer. Coronal tissue sections were cut at 40 μm intervals using a freezing microtome and left free floating for fluorescence immunohistochemistry. Tissue sections were blocked and incubated in phosphate buffer containing 0.3% Triton X-100 (Sigma) and 5% normal donkey serum (Sigma). Incubated on a shaking platform with primary antibodies for 14–18 hr at room temperature. After washing, sections were then incubated for 3 hr with appropriate Alexa Fluor secondary antibodies.

A Leica DMI6000 inverted epifluorescence microscope equipped with Leica DFC365FX monochrome digital camera and Leica LAS-X acquisition software was used for widefield microscopy. For confocal and tile scan confocal imaging, a Leica SP5-II confocal laser-scanning microscope with multi-position scanning stage (Märzhäuser, Germany) was utilized. Primary antibodies used were rabbit anti-mCherry (1:4,000; Biovision), sheep anti-tyrosine-hydroxylase (1:1,000; AB1542, Millipore) and goat anti-ChAT (1:250; AB144P, Millipore). Alexa Fluor 488-conjugated donkey secondary antibodies were used against goat IgG (1:500; Jackson ImmunoResearch) and sheep IgG (1:400; Jackson ImmunoResearch). Alexa Fluor 594-conjugated donkey secondary antibody was used against rabbit IgG (1:1000; Invitrogen).

## Parasympathetic preganglionic model design

A model of the integration of the synaptic drive to the bladder parasympathetic preganglionic neurons in the spinal cord was constructed using NEURON (*Carnevale and Hines, 2006*). The preganglionic neuron was based on using an existing preganglionic neuronal model (*Briant et al., 2014*) that was modified slightly to have a resting potential of ~−65 mV by altering the leak conductance reversal potential. A synaptic drive from Barr[CRH] neurons was modelled by adding an EXPSYN to the soma which was driven with trains of action potentials (20 Hz x 1 s) generated from a NETSTIM to mimic an optogenetic stimulus of Barr[CRH]. The synapse was subthreshold if triggered alone as an input despite a modest degree of summation (~20%) when driven at 20 Hz. A second fast excitatory synaptic drive to model a bladder afferent was added as an EXPSYN to a proximal medial dendrite. This was driven with an incrementing frequency of action potentials (from a NETSTIM based on recordings of pelvic nerve afferents from *Ito et al., 2019*) to model bladder distension-evoked increase in afferent firing over a period of 5 min to represent a typical micturition cycle. The model files are available from data.bris.ac.uk (DOI: 10.5523/bris.20l920gl27ufi204brn8ilonsf).

## Autonomous micturition model design

### Evidence informing the design of model

1. The activation of Barr[CRH] neurons can generate bladder contractions. This is a probabilistic process, with failures. Both the amplitude and the probability of generating an NVC are increased at higher stimulation frequencies. (*Figures 3*, *4* and *11E*)
2. Bladder pressure response to Barr[CRH] drive augments with progress through the micturition cycle. The amplitude of NVCs following identical Barr[CRH] stimulations increases with bladder filling (*Figure 5*) an effect not explained by detrusor muscle stretch (*Figure 5—figure supplement 2*).
3. Barr[CRH] stimulation triggers voids when the bladder is filled (*Figure 5*)
4. Barr[CRH] activity anticipates bladder pressure during the micturition cycle (*Figures 9 and 10*).

5. Barr$^{CRH}$ neuronal excitability does not alter during the micturition cycle (*Figure 11*). Changes in activity in Barr are likely due to greater synaptic drive into Barr$^{CRH}$ neurons, not higher responsiveness to a maintained drive (ie the intrinsic properties of the Barr$^{CRH}$ neurons do not generate the bursts).

6. A spinal gate for Barr$^{CRH}$ drive is opened by bladder distention. This is a hypothesised explanation for point 3, and also explains the experimental observation that the level of Barr$^{CRH}$ activity does not rise linearly with greater sNVC amplitudes (*Figure 11*) (even though as demonstrated in point 1, higher Barr$^{CRH}$ firing rates are capable of producing larger eNVCs, when optogenetic stimulation is used). Also the localisation of this gate at a spinal level is indicated by the response to optoactivation of the descending axons at a spinal level still leading to failures, eNVCs and voids (indicating that this probabilistic gating does not occur in the brainstem, *Figure 6*).

7. The probability of Barr$^{CRH}$ neurons firing at high frequency increases at the end of the micturition cycle. The probe recordings show that the firing of the Barr$^{CRH}$ neurons is only significantly elevated in the final 10–15% of the cycle (*Figures 9E and F and 10B*) and as summarised in the sigmoid relationship (*Figure 10C*).

## Features of the model

Towards the end of the cycle, Barr$^{CRH}$ neurons have a higher probability of increasing their firing rate above baseline levels (as per point 7). We hypothesise that this is due to a greater level of external drive (consistent with point 5) that is via a mechanism which translates bladder pressure into synaptic drive to Barr. This is represented in the model by the first logistic function, marked 'input to Barr' on *Figure 13A* which explicitly replicates the sigmoid seen in the recording data.

The bladder response to bursts of Barr$^{CRH}$ activity consists of a rise in pressure in the shape of a bell curve, of width ~2 s. This represents an NVC and is motivated by points 1 and 4, which show that bursts of Barr$^{CRH}$ activity cause NVCs. The shape of the response is modelled on the profile of responses seen in the recordings. These NVCs summate if triggered in quick succession (as seen towards the end of each micturition cycle).

The amplitude of the NVC is set by the second logistic function (marked 'spinal modulation' on *Figure 13A*). This function generates the increased response to identical levels of Barr$^{CRH}$ firing as the baseline bladder pressure increases with progression through the micturition cycle (points 2 and 6). Again, these responses only become sizeable at the end of the cycle (*Figure 5*), motivating the logistic shape of the curve.

Together, these logistic functions create a positive feedback loop between pressure and Barr$^{CRH}$ firing which generates a regenerative large Barr$^{CRH}$ burst and hence voiding contraction, closely mirroring the responses seen in the real data.

Voiding is considered to have happened when a pressure of 15 mmHg is reached (the model does not include a specific representation of the external urethral sphincter so needs this arbitrary mechanism).

## Model implementation

The model of autonomous micturition employs an iterative algorithm in which bladder pressure and Barr$^{CRH}$ firing rate are updated in each cycle (assumed to last 1 s). At each time point, bladder pressure is incremented by 0.015 mmHg (modelling a continuous infusion). The level of pressure in the bladder is used to determine the maximum level of Barr$^{CRH}$ firing via a logistic function (modelling an afferent input):

$$s_{max}(t) = f_{min} + \frac{(f_{max} - f_{min})}{1 + e^{-k_s(p(t) - m_s)}}$$

Where $s_{max}$ is the maximum spiking rate, and $f_{max}$ and $f_{min}$ represent the maximum and minimum levels of firing – here set to 25 and 4.3Hz respectively, based on spiking rates seen in unit data recordings. The gradient and mean were set to $k_s = 4$ and $m_s = 4.5$. Actual spiking rates were then generated probabilistically by sampling from a random uniform distribution with a maximum level set by $s_{max}$.

The change in bladder pressure produced by the firing of Barr$^{CRH}$ neurons was determined by a logistic function modulated by bladder pressure (representing the level of parasympathetic neuron excitability at the level of the spinal cord):

$$\Delta P(t) = \Delta P_{min} + \frac{(\Delta P_{max} - \Delta P_{min})}{1 + e^{-k_p \left( s(t) - m_p \right)}}$$

Where $s$ is the Barr$^{CRH}$ firing rate and $\Delta P_{max}, \Delta P_{min}$ are the maximum and minimum amplitude of bladder contractions, and $k_p = 0.5$ and $m_p = 6$. The maximum change in pressure depends on the current bladder pressure such that:

$$\Delta P_{max} = 6 * \log(p(t))$$

The output of these equations is shown in *Figure 13* and *Figure 13—figure supplement 1*. Bladder contractions are modelled with a Gaussian bump function, of amplitude $\Delta P_{max}$ and variance 2s. These had a duration of 6 timepoints (approximating the characteristics of non-voiding contractions observed experimentally). At the beginning of each cycle, the current pressure is calculated by summing the baseline pressure (including incrementing due to constant filling) with increases in pressure caused by Barr$^{CRH}$ triggered bladder contractions generated in the current and past cycles.

Voiding occurs at a pressure of 15 mmHg, at which point bladder pressure is decremented at 2 mmHg per cycle (as the bladder empties) until a baseline pressure of <0.3 mmHg is reached and the cycle restarts. The model is coded in MATLAB.

In cycles where optogenetic activation of Barr$^{CRH}$ firing was simulated, the firing rate $s$ was set to 20Hz. To simulate intrathecal Astressin, the mean of the logistic curve for $\Delta P(t)$ was reduced to $m_p = 5$ To simulate attenuation of NVCs, the variance in the level of Barr$^{CRH}$ firing was reduced by 80% but without any change in the mean level of firing.

## Data analysis

### Clustering of multiunit data, waveforms, auto/cross correlations and calculation of firing rates

Multiunit data were recorded on a 32-channel silicon probe (NeuroNexus, Model: A1 × 32-Poly3-10mm-25 s-177-A32), and clustered using spike sorting framework 'Kilosort' (*Pachitariu et al., 2016*). The sorting analysis was carried out using the facilities of the Advanced Computing Research Centre, University of Bristol - http://www.bristol.ac.uk/acrc. Manual curation of clusters was performed in 'Phy' (https://github.com/kwikteam/phy-contrib; *Rossant et al., 2019*) in order to select only well isolated units with clear refractory periods, and to remove artefacts. All further analysis of spike trains and cluster characteristics was carried out in MATLAB.

The centre channel of each cluster was defined as the probe channel on which the waveform was recorded with the maximum range. Representative waveforms were extracted on the centre channel for each cluster by sampling 2000 spikes from the group (if the cluster had fewer than 2000 spikes present, all spikes were used (*Figure 8B,D*). Autocorrelations and cross-correlations were calculated by binning spike trains (1 ms bins) and using the MATLAB function 'xcorr' (*Figure 8D*). Smooth firing rates during laser stimulation events were calculated by convolving the spike train with a normalised Gaussian of standard deviation of 10 ms, using the MATLAB function 'conv2'. (*Figure 8C*). Where z-scored firing rates were required, the MATLAB function 'zscore' was used in with binned spike counts.

### Analysis of bladder pressure and spike trains

Bladder pressure and external urethral sphincter EMG were recorded in Spike2 (CED) and analysed in MATLAB. To compare changes in bladder pressure over multiple recordings, bladder pressure was normalised to between 0 and 1 in each recording, where one represents the maximum pressure recorded during the experiment (*Figure 10A*). The times of voids were identified using the MATLAB function 'findpeaks'; correct identification was verified by eye. Where the bladder pressure was split into phases of the voiding cycle, the period between successive voids was split into the required number of equal time intervals (100 phases in *Figure 10A*, 10 phases in *Figure 10B*). Spike counts during these phases were converted to firing rates by dividing by the width of the relevant time window. This enabled calculation of the mean firing rate in each phase of the voiding cycle (over multiple cycles and cells). In *Figure 10A* firing rates were z-scored to enable comparison between multiple voiding cycles recorded in different animals. Where voiding/inter voiding periods were used

(*Figure 10E*), 'voiding periods' were defined as a window of 15 s either side of the peak of bladder pressure during a void; 'intervoid periods' consisted of all remaining times.

Cross correlograms and Pearson correlation coefficients between spike count and bladder pressure were calculated by downsampling both data to a sampling rate of 1 hz (i.e. 1 s bins for the spike count, 1 hz sampling for the bladder pressure), z-scoring and using the MATLAB function 'xcorr' and 'corr'. (*Figure 10D,E*). All curve fitting was carried out using the MATLAB curve fitting toolbox. In *Figure 10C*, data were fitted to the sigmoid relationship $f(x) = a + \frac{b}{1+e^{c(d-x)}}$ where *a,b,c* and *d* were constants to be determined. 95% confidence intervals were then calculated in MATLAB using the 'confint' function.

### Analysis of urethral sphincter EMG
For EMG data shown in *Figure 9*, artefacts of over 50 times the standard deviation for the recording were removed in MATLAB. The data were then RMS filtered using a 5 s moving window and smoothed using the MATLAB function 'movmean' over 1000 samples (a window of 0.32 s).

### Analysis of non-voiding contractions (NVCs)
NVCs were identified in the intervoid periods only (*Figure 11C*). Pressure measurements during these periods were detrended (using the MATLAB function 'detrend') and NVC peaks were detected using the MATLAB function 'findpeaks', using parameters to select only those peaks greater than 0.1 mmHg above the baseline, between 1.5 and 8 s wide, and at least 5 s from any other such peak. These provided well isolated examples of NVCs for analysis. Spike trains for each cell were binned (2 s bin width for the longer timescales shown in *Figure 11C*; 0.5 s bin width for *Figure 11D*) to extract estimates of firing rate around each NVC and to examine the difference in spike counts during 1.5 s windows beginning 6 s and 3 s before each NVC. For shuffled data a vector of random times was generated for each recording (taken from intervoid periods only) and examined in the same way. The number of shuffled NVC times and real NVC times was equal in each recording To create the trace of bladder pressure shown in *Figure 11D*, bladder pressure was extracted during a 10 s window around each identified or shuffled NVC. Values were referenced to the initial value of pressure recorded during the window in order to extract the change in pressure around the NVC. Windowed measurements were then averaged to generate the mean change in bladder pressure around each identified or shuffled NVC.

## Code availability
Custom MATLAB scripts used to analyse the data are available along with example data are at DOI: 10.5523/bris.20l920gl27ufi204brn8ilonsf along with the MATLAB code for the model of autonomous micturition and the Parasympathetic Preganglionic NEURON model.

## Acknowledgements
Funded by US NIH R01 DK098361 (Anthony J Kanai, Marcus J Drake, Christopher H Fry, Anthony E Pickering). ACS is supported by the Wellcome Trust PhD programme in Neural Dynamics (ref. 108899/Z/15). Thanks to Dr Michael Ambler for assistance with histological analysis and Prof Hidemasa Furue for discussions on spinal mechanisms.

## Additional information

### Funding

| Funder | Grant reference number | Author |
|---|---|---|
| National Institutes of Health | R01 DK098361 | Christopher H Fry<br>Anthony J Kanai<br>Marcus J Drake<br>Anthony E Pickering |
| Wellcome | 108899/Z/15 | Anna C Sales |

The funders had no role in study design, data collection and interpretation, or the decision to submit the work for publication.

## Author contributions

Hiroki Ito, Conceptualization, Data curation, Software, Formal analysis, Validation, Investigation, Visualization, Methodology, Project administration, Writing - review and editing; Anna C Sales, Data curation, Software, Formal analysis, Validation, Investigation, Visualization, Methodology, Writing - review and editing; Christopher H Fry, Anthony J Kanai, Resources, Funding acquisition, Writing - review and editing; Marcus J Drake, Resources, Supervision, Funding acquisition, Project administration, Writing - review and editing; Anthony E Pickering, Conceptualization, Resources, Data curation, Software, Formal analysis, Supervision, Funding acquisition, Validation, Investigation, Visualization, Methodology, Writing - original draft, Project administration, Writing - review and editing

## Author ORCIDs

Anna C Sales (iD) https://orcid.org/0000-0001-8585-3763
Anthony E Pickering (iD) https://orcid.org/0000-0003-0345-0456

## Ethics

Animal experimentation: All experiments and procedures conformed to the UK Animals (Scientific Procedures) Act 1986 and were approved by the University of Bristol Animal Welfare and Ethical review body. licence (PPL3003362).

## Decision letter and Author response

Decision letter https://doi.org/10.7554/eLife.56605.sa1
Author response https://doi.org/10.7554/eLife.56605.sa2

# Additional files

## Supplementary files

• Transparent reporting form

## Data availability

The data generated during this study are included either in the manuscript, in supporting files, or in the dataset deposited at the University of Bristol Research Data Repository at https://doi.org/10.5523/bris.20l920gl27ufi204brn8ilonsf.

The following dataset was generated:

| Author(s) | Year | Dataset title | Dataset URL | Database and Identifier |
|---|---|---|---|---|
| Pickering A, Sales A, Ito H | 2019 | Ito, Sales et al 2019 Example BarrCRH recordings and analysis / model code | https://doi.org/10.5523/bris.20l920gl27ufi204brn8ilonsf | University of Bristol Research Data Repository, 10.5523/bris.20l920gl27ufi204brn8ilonsf |

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
