## [Decision Letter]

**Acceptance summary:**

Your study integrates and advances findings about the nature and function of CrH^+^ neurons in Barrington's nucleus that were previously scattered across species and preparations. As you note, micturition is an essential physiological function and micturition controlling circuitry is apparently conserved across species. Your study advances out understanding of this important circuitry.

**Decision letter after peer review:**

Thank you for submitting your article "Probabilistic, spinally-gated control of bladder pressure and autonomous micturition by Barrington's CRH neurons" for consideration by *eLife*. Your article has been reviewed by three peer reviewers, including Bernardo L Sabatini as the Reviewing Editor and Reviewer #1, and the evaluation has been overseen by Ronald Calabrese as the Senior Editor. The following individual involved in review of your submission has agreed to reveal their identity: Rita J Valentino (Reviewer #2).

The reviewers have discussed the reviews with one another and the Reviewing Editor has drafted this decision to help you prepare a revised submission. In recognition of the fact that revisions may take longer than we typically allow, until the research enterprise restarts in full, we will give authors as much time as they need to submit revised manuscripts.

Summary:

The study by Ito et al. augments our understanding of the function of Barr^CRH^ neurons in autonomous micturition. The authors use a combination of optogenetic and chemogenetic manipulations alongside in vivo electrophysiological recordings to build a model of Barr^CRH^ neuron function. All experiments are performed in anesthetized mice, using cystometry to accurately measure bladder contractions and in some cases using EMG to measure external urethral sphincter (EUS) contractions. This approach is useful as it targets the core autonomous spino-bulbospinal circuit rather than the (potentially more complex and less tractable) volitional circuits. The key novel findings of this paper are twofold. First, the effect of Barr^CRH^ activity is state-dependent; despite being equally excitable throughout the micturition cycle, Barr^CRH^ optogenetic activation only produces contractions in the late stages of the micturition cycle. The effect of Barr^CRH^ activity is therefore dependent on the state of the bladder. Second, the authors report that Barr^CRH^ neuronal firing precedes an increase in bladder pressure, suggesting that Barr^CRH^ neurons provide pre-motor drive to the bladder. Generally, the paper is clearly written and the results are nicely presented within the context of previous work in this field.

Essential revisions:

These are all changes suggested for the text and figures – no new experiments required.

1) It would be helpful to define the various micturition metrics upfront (in Figure 1 or 2) using a schematic. It is not immediately clear how each metric is calculated or how dependent or independent they are of one another (e.g. micturition pressure, threshold pressure, basal pressure, infused volume). A good model would be the schematic in Figure 3C.

2) The chemogenetic experiment is lacking an extended baseline control (e.g. how does void latency change over 120 minute time scales as in Figure 2E). Do the authors have the data to provide this baseline?

3) In Figure 4 the authors argue that "the effect of bilateral stimulation on eNVC amplitude was additive rather than synergistic." This claim is not clearly quantified, and appears to only be true at 20Hz. The effect appears sub-additive at all other frequencies. Please clarify.

4) There are certain considerations that need to be mentioned, particularly when comparing to other studies. It seems that all experiments including neuronal recordings were done under urethane anesthesia. This needs to be explicitly stated in the Abstract. If only some experiments were done under anesthesia, this needs to be made clear. And this needs to be stated in the Discussion as a caveat. Species differences should also be mentioned as a caveat as studies of the micturition reflex have been based on cats, mice and rats and anatomical connections should not be assumed to be identical.

5) There are several concerns about the recordings. First the authors need to indicate whether these were done in the anesthetized or unanesthetized state. It is concerning that out of 113 neurons recorded in 3 animals, only 12 were optically identified. How many neurons were optically activated in each of the three animals? Does that mean that in the other experiments that did not involve recordings, few neurons were activated? Did the authors ever examine fos expression or get some other measure of neuronal activation with optogenetic stimulation? The figure does not suggest that the probe is going through the heart of BN but rather a caudal, sparsely populated section of the nucleus and also is recording from medial locus coeruleus neurons. There are 3 different color traces in Figure 8B which you can only tell when you zoom in greatly. The figure shows traces lateral to the probe as yellow (optically activated) that seem to be in the LC. Some of the "CRH-like" cells in this manuscript appear to be in the medial LC and are even more lateral than the green "non-CRH" cells. A figure showing the probe going through a section of BN that is at the mid-level with a lot of BN neurons and not the major part of the LC is needed. In supplementary data, sections from all three recorded animals should be provided.

---

## [Author Response]

Essential revisions:These are all changes suggested for the text and figures – no new experiments required.1) It would be helpful to define the various micturition metrics upfront (in Figure 1 or 2) using a schematic. It is not immediately clear how each metric is calculated or how dependent or independent they are of one another (e.g. micturition pressure, threshold pressure, basal pressure, infused volume). A good model would be the schematic in Figure 3C.

We have provided an additional figure (Figure 1—figure supplement 2) showing the parameters measured and their relationship. This complements the description in the Materials and methods (subsection “Cystometry, Electromyography and distal colonic manometry”).

2) The chemogenetic experiment is lacking an extended baseline control (e.g. how does void latency change over 120 minute time scales as in Figure 2E). Do the authors have the data to provide this baseline?

The baseline control data for chemogenetic inhibition is provided in Figure 2C. There was no change in control mice (no DREADD with CNO dosing) in the volume infused before a void was triggered (equivalent to the inter-void interval at a constant infusion rate) over the period of 120 minutes. This was not repeated for the infuse-stop protocol in Figure 2E but it can be seen that the effect of CNO was starting to reverse after 120 minutes (as shown in Figure 2D and in 3/5 animals) indicating that its action was not a consequence of a deterioration in the animal’s LUT function over time.

3) In Figure 4 the authors argue that "the effect of bilateral stimulation on eNVC amplitude was additive rather than synergistic." This claim is not clearly quantified, and appears to only be true at 20Hz. The effect appears sub-additive at all other frequencies. Please clarify.

We found no evidence for synergistic effects from bilateral compared to unilateral stimulation. The effects of bilateral stimulation were not significantly different (2-way RM-ANOVA with Sidak’s post hoc tests) from the sum of the responses to stimulation of each side alone (see graph in Author response image 1 of mean± SEM and Figure 5—figure supplement 1). As such we think our statement is appropriate.

4) There are certain considerations that need to be mentioned, particularly when comparing to other studies. It seems that all experiments including neuronal recordings were done under urethane anesthesia. This needs to be explicitly stated in the Abstract. If only some experiments were done under anesthesia, this needs to be made clear. And this needs to be stated in the Discussion as a caveat.“Urethane anaesthesia” is now specified in the Abstract. Also specifically stated in each of the relevant sections of the Materials and methods, and added to the Discussion as a caveat: “We used cystometry in anaesthetised mice to examine the role of Barr^CRH^ neurons specifically in the core processes of autonomous micturition in the absence of behavioural influence. […] This remains to be definitively tested and we hypothesise that the role of Barr^CRH^ neurons in autonomous voiding may be best demonstrated in sleeping rather than conscious, behaving mice.”

Species differences should also be mentioned as a caveat as studies of the micturition reflex have been based on cats, mice and rats and anatomical connections should not be assumed to be identical.

The following text has been added to the Discussion: “In this context it is also important to acknowledge that the pattern of micturition varies to a degree across species and humans and cats do not show the same “squirting” behaviour as their EUS relaxes to allow voiding – this likely involves some distinctive neuronal circuitry (Fowler et al., 2008). […] Importantly, lesions or inhibition of Barrington’s nucleus abolishes voiding in multiple species including humans, cats, rats and mice consistent with it being a core circuit component (reviewed in (Verstegen et al., 2017)).”

5) There are several concerns about the recordings. First the authors need to indicate whether these were done in the anesthetized or unanesthetized state.

All of the recordings were made in the anaesthetised state (now explicitly stated in multiple places as noted above) with the exception of the pelvic nerve stimulation experiment which was made in the decerebrate arterially perfused mouse (Figure 5—figure supplement 2).

It is concerning that out of 113 neurons recorded in 3 animals, only 12 were optically identified.

We recorded around 40 identified single units per animal in the dorsal pons – which is probably close to the number that can be independently isolated from a 32-contact probe in such a small brainstem structure (Barr is ~400µm DV dimension). We report 12 opto-identified Barr^CRH^ neurons and a further 32 that are likely CRH^+^ based on their similar pattern of activity. Thus, our recordings indicate that ~40% (46/113) of the cells are CRH^+^ which is in close agreement with the figures of Hou et al., 2016, who estimate 44% of Barr neurons to be CRH^+^. In terms of the proportion that are opto-identified (~30%) we believe that this is to be expected given that some neurons will be too distant from incident light / shadowed by the recording probe or not sufficiently strongly excited to be driven reliably with a short light pulse (we used a stringent criterion for identification).

How many neurons were optically activated in each of the three animals?

There were 8, 3 and 1 Barr^CRH^ neurons opto-identified in the 3 mice. The corresponding figures for Barr^-CRH-like^ neurons were 9, 19 and 4. Now specified in the legend to Figure 10.

Does that mean that in the other experiments that did not involve recordings, few neurons were activated?

As noted above, we think that the number of identified neurons in recordings is an underestimate of the opto-activated population as A) the recording electrode can only ‘see’ a subset of neurons in Barr, B) need for coincident alignment of tracks for separate recording probe and illumination from optical fibre C) ‘shadowing’ of some cells from light by the recording electrode D) strict criteria for both initial identification discrete single units and subsequent opto-identification (meaning some Barr^CRH-like^ neurons that were weakly excited were excluded from the Barr^CRH^ subset). We also note that the magnitude of our bladder responses to Barr^CRH^ opto-activation were similar to that reported in the work of Hou and colleagues (Hou et al., 2016) suggesting that we were opto-activating a similar proportion of neurons.

Did the authors ever examine fos expression or get some other measure of neuronal activation with optogenetic stimulation?

We did not try cfos or any other measure beyond the important functional measure of the bladder responses to opto-activation. This was used as a reliable marker indicating the proximity of the optical fibre to Barrington’s nucleus.

The figure does not suggest that the probe is going through the heart of BN but rather a caudal, sparsely populated section of the nucleus and also is recording from medial locus coeruleus neurons.

We have shown the section showing the best combination of features from a 1:3 series of 40µm sections through Barr. This means we get only 2-3 sections containing Barr neurons. In this particular case the probe track was most obvious in this section and so it was selected for illustration purposes (always something of a compromise between the best anatomy, clearest demonstration of the location of the probe, etc) – we think it provides a noteworthy juxtaposition of probe recording site, the distribution of opto-identified cells and labelled neurons. We believe that the location of the track will have allowed recordings from neurons in the core of Barr (its recording surface was oriented to face rostrally) and certainly this is borne out by the close similarity between our extracellular recordings and those in other papers using fibre photometry to sample the population signal of Barr^CRH^ neurons (Hou et al., 2016, Keller et al., 2018 and Verstegen et al., 2019).

There are 3 different color traces in Figure 8B which you can only tell when you zoom in greatly. The figure shows traces lateral to the probe as yellow (optically activated) that seem to be in the LC. Some of the "CRH-like" cells in this manuscript appear to be in the medial LC and are even more lateral than the green "non-CRH" cells.

Our figure has inadvertently misled the reviewers. The position of the spikes refers to their clustering onto the probe recording sites (the site with the strongest signal) rather than their specific location within the brain section as shown. We definitely did not transduce the LC with the opsin in these experiments (hence none of the opto-identified neurons can be from the LC). However, we agree that we may have made some recordings from LC neurons that could conceivably have been included in the Barr^CRH-like^ neurons or the non-identified category. Therefore, to test our characterisation with respect to the Barr^CRH-like^ vs. LC neurons, we have included new data showing recordings from opto-identified LC neurons in mice (n=3) that had been selectively transduced with ChR2 using CAV-PRS-ChR2-mCherry (Li et al., 2016). We used an identical recording configuration except the recording probe was introduced 100µm lateral to our usual recording position for Barrington’s nucleus. These opto-identified mouse LC neurons (n=29) showed the characteristic pattern of tonic activity with a phasic response to pinch. Interestingly they also showed a phase locked activation during voiding (like that reported in rats by Manohar et al., 2017) whose magnitude was significantly smaller than that seen in either Barr^CRH^ or in Barr^CRH-like^ neurons but is larger than that seen in the non-identified population. We have included this LC data as a supplementary figure (Figure 10—figure supplement 1) and revised the manuscript (subsection “Barr^CRH^ activity anticipates bladder pressure during the micturition cycle”, last paragraph) to support our identification of these distinct neuronal groups.

A figure showing the probe going through a section of BN that is at the mid-level with a lot of BN neurons and not the major part of the LC is needed. In supplementary data, sections from all three recorded animals should be provided.

We chose the histological section that we felt best illustrated the position of the recording probe and its relationship to labelled cells and would prefer to retain this figure. However, we have included another image from this mouse to help convince the reviewers that the probe track was also evident in a section at the mid-level (#94) but this section was of lower quality and so we chose not to use it. We also include a histological image of pontine tissue from another recording (#83) with the location of the track. We confirmed the location of the third recording (#80) as being within Barr but those sections were damaged in processing (torn at the recording site) and unfortunately they were not imaged (as they were not felt to be of publication quality). We do not think it adds much value to the reader to add these images to a supplementary figure, but they could be included in the transparent review process for the interested.

**Author response image 2. respfig2:**